



# Subglacial lakes and hydrology across the Ellsworth Subglacial Highlands, West Antarctica

Felipe Napoleoni[1], Stewart S. R. Jamieson[1], Neil Ross[2], Michael J. Bentley[1], Andrés Rivera[3], Andrew M. Smith[4], Martin J. Siegert[5], Guy J. G. Paxman[1, 6], Guisella Gacitua[7], José A. Uribe[8], Rodrigo Zamora[8], Alex M. Brisbourne[4], and David G. Vaughan[4]

[1]Department of Geography, Durham University,Durham, UK
[2]School of Geography, Politics and Sociology, Newcastle University, Newcastle upon Tyne, UK
[3]Departamento de Geografía, Universidad de Chile, Santiago, Chile
[4]British Antarctic Survey, Cambridge, UK
[5]Grantham Institute and Department of Earth Science and Engineering, Imperial College London, London, UK
[6]Lamont-Doherty Earth Observatory, Columbia University, Palisades, New York, USA
[7]Instituto de Ciencias Físicas y Matemáticas, Universidad Austral de Chile, Valdivia, Chile
[8]Centro de Estudios Científicos, Arturo Prat 514, Valdivia, Chile

**Correspondence:** Felipe Napoleoni (felipe.a.napoleoni@durham.ac.uk)

**Abstract.** Subglacial water plays an important role in ice sheet dynamics and stability. It is often located at the onset of ice streams and has the potential to enhance ice flow downstream by lubricating the ice-bed interface. The most recent subglacial lake inventory of Antarctica mapped nearly 400 lakes, of which ∼14% are found in West Antarctica. Despite the potential importance of subglacial water for ice dynamics, there is a lack of detailed subglacial water characterization in West Antarctica.

Using radio-echo sounding data, we analyse the ice-bed interface to detect subglacial lakes. We report 37 previously uncharted subglacial lakes and present a systematic analysis of their physical properties. This represents a ∼ 60% increase in subglacial lakes in the region. Additionally, a new digital elevation model of basal topography was built and used to create a detailed hydropotential model of Ellsworth Subglacial Highlands to simulate the subglacial hydrological network. This approach allows us to characterize basal hydrology, subglacial water catchments and connections between them. Furthermore, the simulated

subglacial hydrological catchments of Rutford Ice Stream, Pine Island Glacier and Thwaites Glacier do not match precisely with their ice surface catchments.

## 1 Introduction

Subglacial water is important for ice sheet flow, with the potential to control the location of ice stream onset (e.g., Siegert and Bamber, 2000; Vaughan et al., 2007; Winsborrow et al., 2010; Wright and Siegert, 2012) by lubricating the ice base and

reducing basal friction (Bell et al., 2011; Pattyn, 2010; Pattyn et al., 2016; Gudlaugsson et al., 2017). Some studies have reported acceleration of ice velocity in different regions of Antarctica as a result of basal hydrologic conditions (e.g., Stearns et al., 2008). Subglacial water piracy has been invoked to explain the on and off switching of streaming flow (e.g., Vaughan et al., 2008; Anandakrishnan and Alley, 1997; Diez et al., 2018). Additionally, small changes in the ice sheet surface or ice



thickness can lead to significant changes in basal hydrology; causing water flow to change direction (Wright et al., 2008).

Significant glaciological change is known to have taken place in West Antarctica over the last few thousand years (Siegert et al., 2004b, 2019). For example, there is evidence of a well-organized and dynamic subglacial hydrological system which formed paleo-channels and basins underneath the present Amundsen Sea Embayment (Kirkham et al., 2019). This subglacial hydrological system was most likely caused by episodic releases of meltwater trapped in upstream subglacial lakes (Kirkham et al., 2019). But these changes have not been uniform across WAIS. Ross et al. (2011) demonstrated that the ice divide and the

ice flow across the Ellsworth Subglacial Highlands (ESH) have been stable for more than 20 ky. The ESH are located within the Ellsworth-Whitmore Mountain block (Figure 1). Some studies have demonstrated the potential variability of subglacial flow routing and many that subglacial lakes form part of a highly dynamic drainage network (e.g., Siegert, 2000; Fricker et al., 2014; Pattyn et al., 2016). However, our understanding of the subglacial hydrology in ESH is limited: only 2 subglacial lakes (i.e., subglacial lake Ellsworth and CECs) have been recognized within the troughs of this region. Understanding the

current hydrological network, and assessing its evolution and sensitivity through time, is therefore essential for an improved understanding of Antarctic ice-sheet dynamics. Additionally, a better comprehension of this relationship is also important for studies of ice sheet mass balance and supplies of water to the ocean potentially affecting circulation and nutrient productivity (Ashmore and Bingham, 2014).

The most recent inventory identified ∼400 subglacial lakes across Antarctica (Wright and Siegert, 2012, Figure 1), ∼14%

of which are located beneath the West Antarctic Ice Sheet (WAIS). Studies have shown that some of these subglacial lakes are connected (Wingham et al., 2006; Fricker et al., 2014) and that some are highly dynamic, and drain and refill (e.g., Fricker et al., 2007, 2014). These active subglacial lakes have been identified using a range of techniques including satellite measurements of ice surface elevation changes (e.g., Wingham et al., 2006; Smith et al., 2009), characterisation of the subglacial topography from ice surface data (e.g., Bell et al., 2007; Bell, 2008; Jamieson et al., 2016); airborne radio echo sounding (RES) (e.g.,

Robin et al., 1970; Popov and Masolov, 2003); and/or ground-based RES (e.g., Rivera et al., 2015).

Previous work in the ESH area identified Subglacial Lake Ellsworth (SLE) and Subglacial Lake CECs (SLC) (Figure 1) by interpreting specular basal reflections in RES data as an indicator of deep (>10 m) subglacial water (e.g., Siegert et al., 2004a; Rivera et al., 2015). Subglacial Lake Ellsworth's water depth, geometry and lake floor sediments were characterised using seismic reflection surveys (Woodward et al., 2010; Smith et al., 2018). SLE and SLC are components of a subglacial

hydrological network in the upper reaches of multiple West Antarctic ice streams and in the ESH (e.g., Vaughan et al., 2007). However, despite the evidence of subglacial water, and a potential subglacial network connecting multiple subglacial water bodies, this region is not fully understood in terms of subglacial hydrological dynamics. Given the fact that this region is located up-ice of the fastest-changing ice streams in the world (e.g., Pine Island Glacier and Thwaites Glacier), and that they are some of the most vulnerable glaciers to ongoing climate change (Martin et al., 2019), a more detailed study of the subglacial

hydrological system using existing RES data is of particular importance. Our aim is to produce an inventory of subglacial lakes for the ESH, and to model the modern subglacial hydrology in the ESH draining towards the Amundsen Sea Embayment. We then assess the connectivity of these new subglacial lakes and the potential drainage flow of the ice basal water to the edge of the continent.

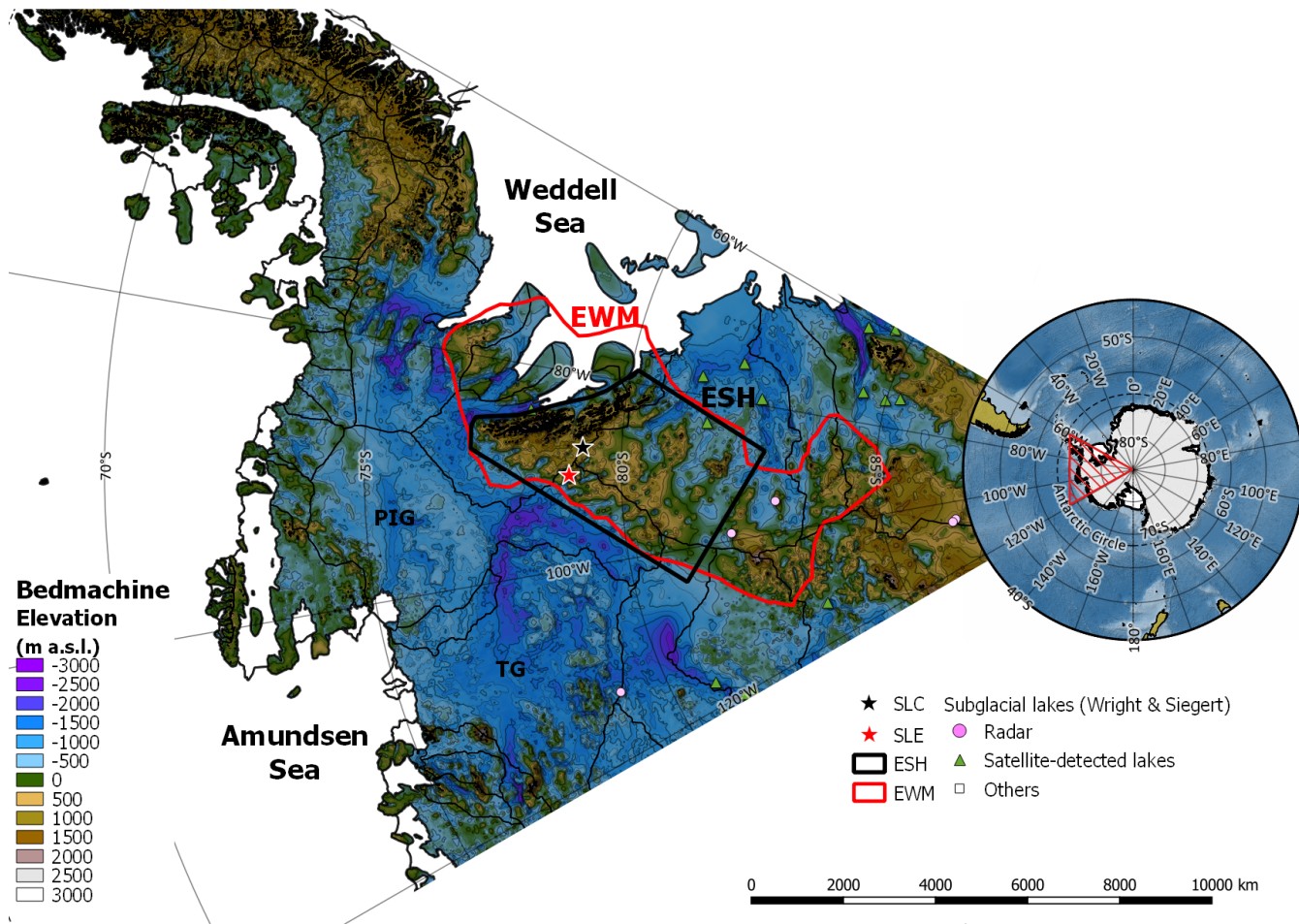

**Figure 1.** The distribution of West Antarctic subglacial lakes between 60°W and 120°W (Wright and Siegert, 2012). EWM: Ellsworth-Whitmore Mountain (in red, polygon from Jordan et al. (2013)); ESH: Ellsworth Subglacial Highlands (in black). In the background is Bedmachine elevation model (500 m) (Morlighem et al., 2019), with contour lines every 500 m. Projection: Antarctic Polar Stereographic (EPSG 3031).

## 2 Methods

During the 2004/2005 austral summer the British Antarctic Survey collected ∼ 35, 000 km of airborne RES data, mostly over the catchment of PIG (Vaughan et al., 2006) (Figure 4). The survey aircraft was equipped with dual-frequency carrier-phase GPS for navigation, a radar altimeter for surface mapping, magnetometers and a gravimeter for potential field measurements and the Polarimetric radar Airborne Science INstrument (PASIN) ice-sounding radar system (Vaughan et al., 2006, 2007; Corr et al., 2007). The radar system was configured to operate with a transmit power of 4 kW around a central frequency of 150

MHz. A 10-MHz chirp pulse was used to successfully obtain bed-echoes through ice more than 4200 m thick (Vaughan et al.,



2006). Here, we use the radar data processed as a combination of coherent and incoherent summation without SAR processing
to obtain ice-bed interface information (Vaughan et al., 2006). We analyse a subset of this data-set to characterize the bed
conditions of the northern margin of the ESH. We focus on three main tasks: first, identifying subglacial water at the ice base;
second, defining and characterizing the modern subglacial hydrological network; and third, simulating the subglacial flow

routing. We identify subglacial lakes by analysing the power of the reflected energy from the ice-bed interface (Bed Reflection
Power or BRP, Gades et al. (2000)) using four steps. We use the radar data to identify bright reflections underneath the ice
(Section 2.1) and then we correct the ice attenuation of the radar power to obtain absolute reflections (Section 2.2). We then
classify the identification confidence of each potential water body (Section 2.4) according to the methodology of Carter et al.
(2007). Lastly, we analyse the hydraulic potential and ice surface slope of each bright reflection (Section 2.6).

## 70  2.1   Identification of subglacial lakes

Nearly 7500 km of the BBAS airborne radar (i.e., flight lines: B01, B02, B03, B05, B08, B09, B22, T04) data were analysed
to identify high amplitude basal reflections potentially associated with a subglacial lake signature. A preliminary qualitative
approach allowed us to identify a variety of bright surfaces at the base of the ice sheet. These reflections were then quantitatively
analysed to determine the BRP in order to classify them as potential subglacial lake candidates, saturated sediments and/or

smooth surfaces following previous methods (e.g., Robin et al., 1970; Oswald and Robin, 1973; Popov and Masolov, 2003;
Siegert, 2005; Rivera et al., 2015). We looked for lake candidates that satisfy the following five criteria:

1. Lake surfaces must be smooth and planar (Kapitsa et al., 1996; Siegert, 2005).

2. The potential water surface should incline at $\sim 11$ times opposite to the ice surface slope (Oswald and Robin, 1973).

3. There should be a constant hydrological low along the length of the lake (Vaughan et al., 2007).

4. The lake should have BRP values that are significantly higher than the surrounding surface return (15-20dB) Siegert
(2000).

5. There should be a low amplitude strength variation (specularity) of the ice-water interface (i.e. $< 3\sigma$ BRPr) (Carter et al.,
2007).

## 2.2   Bed Reflection Power (BRP)

To distinguish the potential lakes from their surroundings, the normalized depth-corrected BRP was computed for the BBAS
data over each candidate lake, following previous studies (e.g., Gades et al., 2000; Jacobel et al., 2009; Matsuoka et al., 2012;
Schroeder et al., 2016; Young et al., 2016), as the ratio of the locally measured power of the basal echo ($BRP_{\mathbf{m}}$) to the energy
estimated from the ice thickness and the empirical fit ($BRP_{\mathbf{e}}$):

$$BRP = \frac{BPR_{\mathbf{m}}}{BRP_{\mathbf{e}}} \tag{1}$$

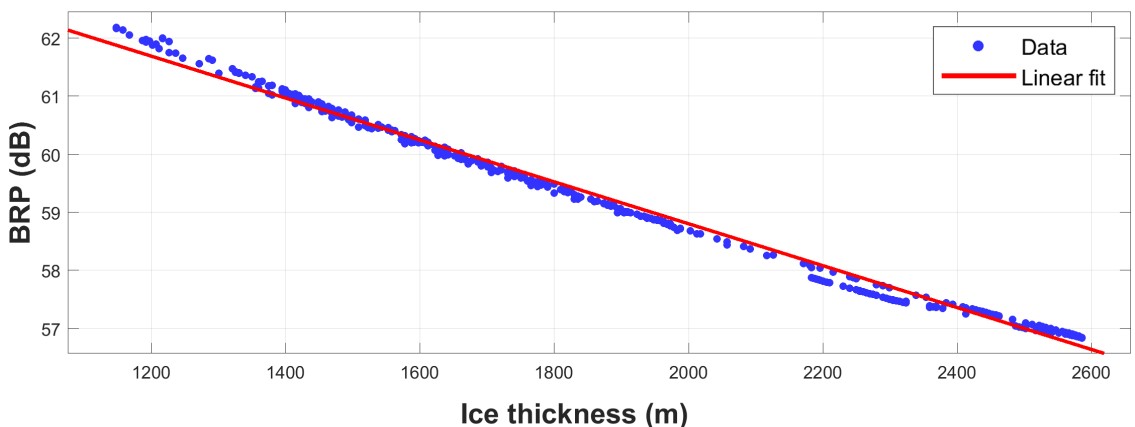

**Figure 2.** Example of the ice-attenuation correction for a section of PASIN radar data (BBAS line B01: SL2 in Fig.6). linear fit: the red line is a log of normalized power representing the BRP dependency to ice thickness.

In calculating the $BRP$ we account for the geometrical spreading ($L_\mathbf{t}$), attenuation loss and system gains ($L_\mathbf{i}$) encountered during the survey. In addition, we account for the height of the airborne BBAS system over the ice surface and for the ice thickness. The $BRP_\mathbf{e}$ quantification was adopted from previous work (e.g., Bentley et al., 1998). A simplified version of the model can be written as follows:

$$BRP_\mathbf{e} = (P_\mathbf{tx} + G_\mathbf{a} + G_\mathbf{ant}) - (L_\mathbf{g} + L_\mathbf{i} + T_\mathbf{ai}) \qquad (2)$$

Here $P_\mathbf{tx}$ is the power emitted by the radar energy source, $G_\mathbf{a}$ corresponds to the gain of the receiver without including the antenna gain; $G_\mathbf{ant}$ is the gain of the antenna (Heliere et al., 2007), and $T_\mathbf{ai}$ is the transmission coefficient at the first interface (air-ice). $L_\mathbf{g}$, is the geometrical spreading loss; $L_\mathbf{i}$, is the ice attenuation loss. These latter parameters are modelled separately and substituted into (2) for each subglacial lake candidate.

The geometrical spreading loss ($L_\mathbf{g}$) is calculated after Bentley et al. (1998) as:

$$L_\mathbf{g} = \frac{(G_\mathbf{ant}\lambda)^2}{[8\pi(h_\mathbf{a} + \frac{h_\mathbf{i}}{\varepsilon_\mathbf{i}})]^2} \qquad (3)$$

where $G_\mathbf{ant}$ is the antenna gain (11 dBi), $\lambda$ is the wavelength of the radar signal at the air medium (1.935 m), and $h_\mathbf{a}$ is the height of the antenna (values taken from Corr et al. (2007)). $h_\mathbf{i}$ is the ice thickness, derived from BBAS ice thickness picks, and $\varepsilon_\mathbf{i}$ is the relative electric permittivity of ice (3.188) (Glen and Paren, 1975).

For each section of radar profile, we compute the ice attenuation loss, calculating a depth-averaged attenuation rate. In this
approach we used an empirical relationship between the ice thickness ($Z_\mathbf{s}$) and the BRP, and then normalized the received power to a constant depth (Jacobel et al., 2009) (Figure 2).



## 2.3 Specularity calculation

The standard deviation ($\sigma$) of the BRP was used to determine how specular the surface of each potential water body is (Carter et al., 2007). This was used as a threshold to determine whether the surface was smooth or rough (Peters et al., 2005). The standard deviation ($\sigma$) was calculated within 40 samples around a particular point to ensure a representative number of radar observations are included. A low value of $\sigma$ indicates a smooth surface is present at the base of the ice as would be expected for the surface of a water body (Rivera et al., 2015; Gacitúa et al., 2015; Carter et al., 2007; Bowling et al., 2019).

## 2.4 Subglacial Lake classifications

Having identified the candidate subglacial lakes, we determine the degree of confidence in our identification by ranking them from the most to least likely to be a subglacial lake. To achieve this we initially use the BRP values for each lake candidate following previous work (Carter et al., 2007), comparing these values to already known subglacial water bodies such as Subglacial Lakes Ellsworth, CECs and Vostok. This leads to a suite of four physically-based categories to which each potential subglacial lake is assigned, ranging from definite to indistinct as follows:

1. **Definite subglacial lakes**. This category has an absolute reflectivity higher than the surroundings (BRPr 15db higher) and displays a low variation in the BRPr (high specularity: $< 3\sigma$ BRPr). Therefore, this category is defined by a high absolute reflection power, a low standard deviation ($\sigma$) across the subglacial lake candidate surface, and a flat hydraulic surface.

2. **Dim subglacial lakes**. These candidates have a high relative reflection strength and surface specularity, but lack the absolute reflectivity values of definite lakes in that their BRPr is no more than 10db higher than the surroundings.

3. **Fuzzy subglacial lakes**. These candidates show higher relative and absolute reflection coefficients than the surroundings, but are not specular along their surfaces (i.e. $> 3\sigma$ BRPr). We note that a challenge with such candidates is that they could also potentially be interpreted as saturated basal sediments (Dowdeswell and Siegert, 2003; Peters et al., 2005; Siegert, 2000). In addition, if the water is less than 8 m deep, reflections from the water-lake bottom interface may interfere with the signal from the ice-water interface (Gorman and Siegert, 1999). Furthermore, exceptionally smooth surfaces with no water present could also have similar signal characteristics to these fuzzy subglacial lakes (Carter et al., 2007).

4. **Indistinct subglacial lakes**. This category is composed of lake candidates that are specular but are difficult to distinguish from the surroundings. Although these could still represent subglacial lakes, such characteristics are also common to transient water systems or to exceptionally smooth beds with fine grained sediments surrounded by rougher saturated sediments (Carter et al., 2007).





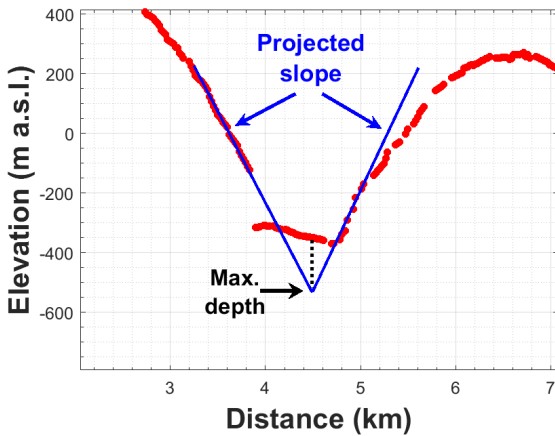

**Figure 3.** Example of subglacial water depth calculation in BBAS line: B02: SL9 (Figure 7b). Red dots show the RES-detected spatial extent of the subglacial interface, the horizontal portion represents the subglacial water surface. The blue dashed line over the red points is the extrapolation of the side-slope topography and the maximum depth is observed where both side-slopes converge.

## 2.5 Subglacial Lake dimensions

To measure the size of the water bodies we calculated the length of the bright reflector in each radargram associated with a potential subglacial lake. We then selected each trace within the potential subglacial lake window and visualised it in a GIS to examine its spatial extent. We consider the spatial extent as being either a minimum or maximum lake length since we do not know the full 2-dimensional shape of the water body due to the reconnaissance nature of the airborne survey. For each subglacial lake we then estimated the area in two different ways:

1. Assuming a circular shape with a radius equal to half of the lake length.

2. Assuming an elliptical shape and using the length of the trace as either the width (a) or the length radius (b). We then use SLE and SLC, as trough-confined examples of subglacial lakes, to obtain an average of the a:b ratio (i.e., ~4:1) and we calculate the area in each case.

Using the method proposed by Dowdeswell and Siegert (1999), we estimated the depth by extrapolating the observed side-slope topography at the margin of each subglacial lake (Figure 3).

## 2.6 Hydropotential ($\Phi$) calculation

To determine the subglacial hydrological characteristics of the Ellsworth-Whitmore Mountain (EWM) region we produced a new bed Digital Elevation Model (DEM). We use existing gridded bed elevation data from: Bedmap2 (Fretwell et al., 2013); ice thickness measurements from DELORES (2007-2009) and CECs (2005/2006) (Siegert et al., 2012); new along-track ice thickness measurements from the 2014 CECs RES campaign (Rivera et al., 2015); and new unpublished radar measurements





from the CECs 2017 RES field campaign in ESH region (Zamora et al., 2019; Uribe et al., 2019). We use a 2 km grid mesh
with a continuous curvature tension spline algorithm (Paxman et al., 2017; Wessel et al., 2013) to grid the data. We masked
the grid to remove interpolated values more than 5 km from the nearest measured data point, and replaced them with Bedmap2
bed elevation values. We also masked the ice shelves from the radar measurements and replaced these values with those from
Bedmap2 to obtain offshore bathymetry. The 1 km resolution CryoSat2 ice surface DEM (Slater et al., 2018) was down-
sampled to 2 km to match the new bed DEM. This enabled us to use the ice surface elevation and subglacial bed to determine
the hydrological head, $\Phi$, following Shreve (1972).

At the ice sheet bed, water flows in the direction of steepest descent of the hydraulic potential. Hydropotential ($\Phi_{\mathbf{h}}$) is the
sum of the water pressure, $P_{\mathbf{w}}$, and water density, $\rho_{\mathbf{w}}$ (kg m$^{-3}$), normalized by gravitational acceleration ($\mathbf{g}$) and the water
system bed elevation, $Z$ (m) (e.g., Shreve, 1972; Cuffey and Paterson, 2010; Livingstone et al., 2013) as follows:

$$\Phi = P_{\mathbf{w}} + \rho_{\mathbf{w}}\mathbf{g}Z \tag{4}$$

Equation 4 can usefully be rewritten into an alternative equation in terms of subglacial bed and ice surface elevation (Shreve,
1972).

$$\Phi = \rho_{\mathbf{w}}gZ + k\rho_{\mathbf{i}}gH \tag{5}$$

where $\rho_{\mathbf{i}}$ is the density of ice (kg m$^{-3}$), $H$ is the ice thickness (m) and $k$ is a dimensionless factor, representing the influence
of ice overburden pressure on the local subglacial water pressure.

Assuming the water pressure is close to the ice overburden pressure ($k \sim 1$) from seismic and borehole observations (e.g.,
Blankenship et al., 1986; Tulaczyk et al., 2001), equation 5 can be rearranged as follows:

$$\Phi = \rho_{\mathbf{w}}gZ + \rho_{\mathbf{i}}gH \tag{6}$$

This approach has been widely used across Antarctica to model the subglacial hydrological drainage (e.g., Livingstone et al.,
2013; Carter et al., 2017; Kirkham et al., 2019).

### 2.7 Subglacial water flow routing

Since subglacial water tends to move from areas of high to low subglacial water pressure, following the hydropotential ($\Phi$)
gradient (Shreve, 1972), we can determine present-day large-scale subglacial flow routing and identify whether the candi-
date subglacial lakes connect into this subglacial hydrological network. We modelled the subglacial flow routing using the
hydropotential ($\Phi$) and followed Schwanghart and Scherler (2014) to calculate the flow routing using the following steps:

1. Lows (sinks) in hydropotential ($\Phi$) were filled to their lowest pour point.

2. The channelized network was then determined using a multiple flow direction (MFD) algorithm (Schwanghart and
   Scherler, 2014). The subglacial hydraulic drainage basin was then delineated, and we computed the flow accumulation
   in order to understand the upstream contributing area above the lakes.





3. The stream network was then defined using an up-slope area threshold for channel initiation. We set an arbitrary threshold of 50 connected cells (100 km$^2$) as defining a channel.

## 3   Results

### 3.1   Subglacial lakes

Using qualitative viewing of the BBAS radar data, we identified 107 bright reflections potentially caused by subglacial lakes within the BBAS dataset (Figure 4). These reflections were further analysed by comparing the characteristics of these features with other Antarctic subglacial lakes (Siegert, 2005; Carter et al., 2007) in order to either confirm or reject each feature as a subglacial lake (Figure 5).

The lakes are largely distributed within a series of subglacial valleys that emerge from the ESH into the Bentley Subglacial Trench near the ice divide between Pine Island Glacier (PIG), Rutford Ice Stream (RIS) and Institute Ice Stream (IIS) at the northern edge of the ESH (Figure 6). We observe two clusters of subglacial lakes and one potential chain of subglacial lakes in the EWM region. The first cluster is near SLE, <100 km from the ice divide between IIS, RIS and PIG (Figure 7a). Most of the subglacial lakes in this first cluster are located in the same trough as SLE, or are connected to the same trough system. However, in some cases the drainage of subglacial water in this cluster may be in two distinct directions (i.e., towards Weddell Sea Embayment or Amundsen Sea Embayment, Figure 7c). The second cluster is located in a valley upstream of the PIG catchment near the water divide between Amundsen and Weddell Sea (Figure 7b). In this cluster, the hydraulic modelling suggests the lakes connect and drain into the modern hydrological network flowing towards the Weddell Sea (Figure 7d). The chain of subglacial lakes is in a trough located upstream of IIS and PIG, less than 50 km from the ice divide. These subglacial lakes are in between Ellsworth Trough and the valley that hosts the second cluster of lakes, near the PIG-IIS ice divide (Figure 7b). Furthermore, the subglacial hydrological simulation also shows that part of the drainage flowing beneath the PIG is diverted to flow beneath Thwaites Glacier, collecting the water of these ice catchments and draining to the Amundsen Sea Embayment.

Using the quantitative analysis of the bed reflectivity at 107 sites, and classifying accordingly, we confirm the presence of 37 previously unrecognised subglacial lakes (Supplementary information: Table 1), which is a ∼60% increase in the total number of subglacial lakes known to exist beneath the WAIS (Wright and Siegert, 2012). Although a small number of these subglacial lakes were hypothesised or identified by other studies (e.g., Livingstone et al., 2013; Vaughan et al., 2007), none of these water bodies were included in the most recent subglacial lake inventory of Wright and Siegert (2012). Using the categories described above, we categorise these subglacial lakes into 4 groups with different confidence levels (Figure 6). We identify 16 definite subglacial lakes (very high confidence), 13 dim subglacial lakes (high confidence), 3 fuzzy subglacial lakes (medium confidence) and 3 indistinct subglacial lakes (low confidence).



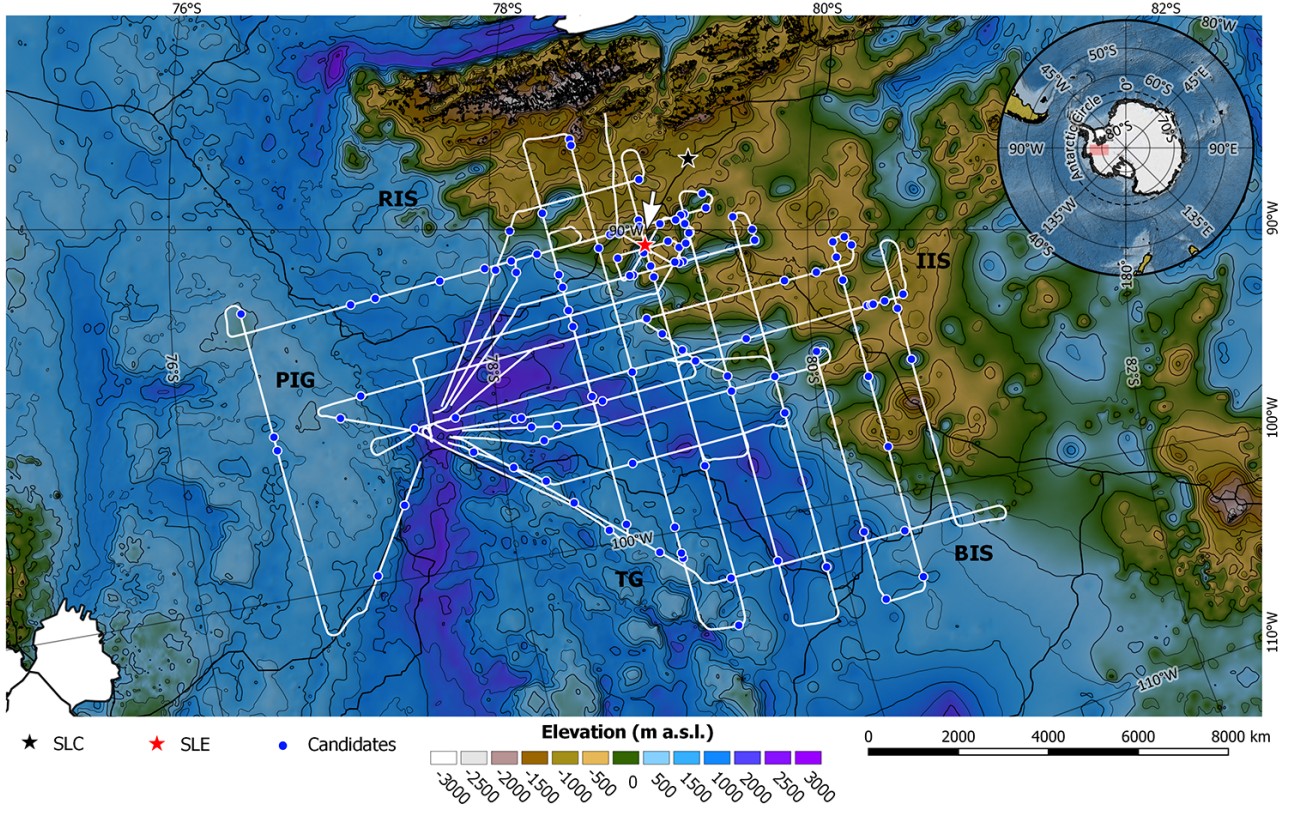

**Figure 4.** Qualitative analysis of high amplitude reflection (blue circles) identified using eight flight lines of the BBAS radar set. The black and red star show the position of Subglacial Lake CECs (SLC) and Subglacial Lake Ellsworth (SLE), respectively. The red star also marks the radar section shown in Figure 5. Data from the BedMachine Elevation model (500 m) (Morlighem et al., 2019) is shown in the background. White lines show the 7500 km of analysed BBAS radar lines. Black lines: catchment boundaries are from Fretwell et al. (2013). RIS: Rutford Ice Stream; PIG: Pine Island Glacier; TG: Thwaites Glacier; BIS: Bindschadler Ice Stream; IIS: Institute Ice Stream. Projection: Antarctic Polar Stereographic (EPSG 3031).

## 3.2 Distribution of subglacial lakes


The ESH hosts 28 of the new subglacial lakes, 7 others are in the Bentley Subglacial Trench, and 2 lakes are located in the region of high topography between tributaries 3 and 5 of PIG, between Byrd Subglacial Basin and the outlet of PIG (SL 28 and 29 in Figure 6). The majority of these subglacial lakes (34) are very close to an ice divide and most of them (15) are located within 20 km from the ice divide between IIS and RIS (Figure 8a). Most of the small (less than 3 km length) subglacial lakes

(17) lie within 50 km of an ice divide, and whereas the largest subglacial lakes (SL 28-27-19) are all located within 25 km of the PIG ice divide. The ice thickness over these subglacial lakes is variable and ranges from 1600 m to 4000 m with an average thickness of ∼ 2600 m (Figure 8b). Few subglacial lakes (13) are covered by ice thicker than 3000 m thick and even fewer (7) lie underneath thinner ice (1500-2000 m thick). Most subglacial lakes associated with thinner ice columns are located

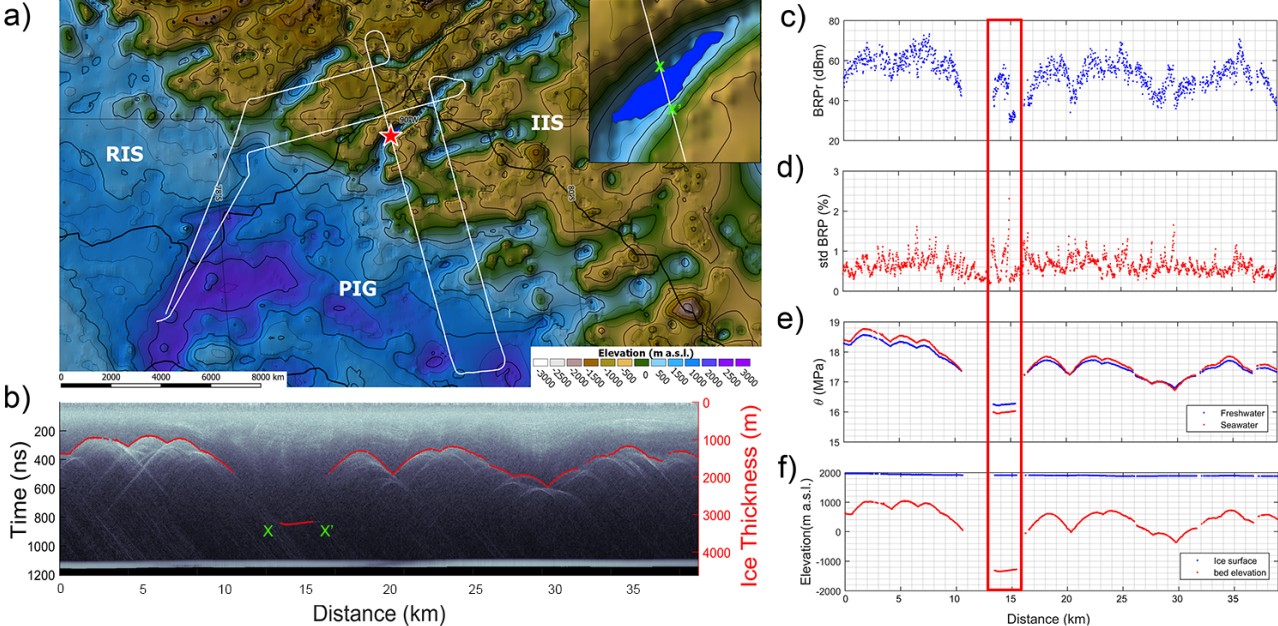

**Figure 5.** Example of subglacial lake identification in a radargram from the BBAS survey (Flight line B05) and its location in WAIS. a) Shows the location of the flight line (B05, white-line) and the position of the radargram (red star) in the line overlain on the new subglacial DEM (2 km). The inset on 'a' shows the Subglacial Lake Ellsworth bed topography contour, and the portion highlighted in green represents the subglacial lake in 'b'. RIS: Rutford Ice Stream; IIS: Institute Ice Stream; PIG: Pine Island Glacier. Projection: Antarctic Polar Stereographic (EPSG 3031). b) Shows the bright reflection (red line) classified as a subglacial lake. The radargram is corrected for elevation and shows both the time (ns) for the returned echo from the ice/bed interface (Y1) and the ice thickness (m) calculated using a radio wave propagation through the ice of 0.168 m ns$^{-1}$ (Y2). c) Bed Reflection Power (dBm). d) Specularity ($\sigma$BRP). e) Hydropotential considering fresh (blue) and salt (red) water densities. f) Ice surface elevation (blue line) and bed elevation (red line).

south of the IIS-PIG ice divide, while those underneath thicker ice columns are distributed at the head of PIG. The majority of
subglacial lakes with an overlying ice thickness of between 2000 and 3000 m are situated close to the triple ice divide between
IIS-RIS and PIG and along the border between PIG and RIS, where the ice surface slope is near zero and, hence, the subglacial
hydraulic gradient is also close to zero. The length of subglacial lakes ranges from a minimum of $\sim 0.35$ km to a maximum
of $\sim 22$ km with a mean of $\sim 4$ km (Figure 8c). In general, ice surface velocities over subglacial lakes is never higher than 60
myr$^{-1}$ and most of the subglacial lakes (25) lie beneath ice flowing less than 6 myr$^{-1}$ (Mouginot et al., 2019). The location of
the newly discovered subglacial lakes and the distribution of ice surface velocities are shown in Figure 9.

The new bed DEM of the ESH provides new detail on two key subglacial troughs and shows that they are much deeper
than shown in existing DEMs (e.g. Bedmap2 and BedMachinev1). These troughs extend parallel to the Ellsworth Mountains
which faces the Weddell Sea and are extensive linear features that appear to connect the interior of the ESH to the deep basins
that lie beneath the WAIS. Additionally, the DEM reveals that subglacial lakes are hosted within two different subglacial to-

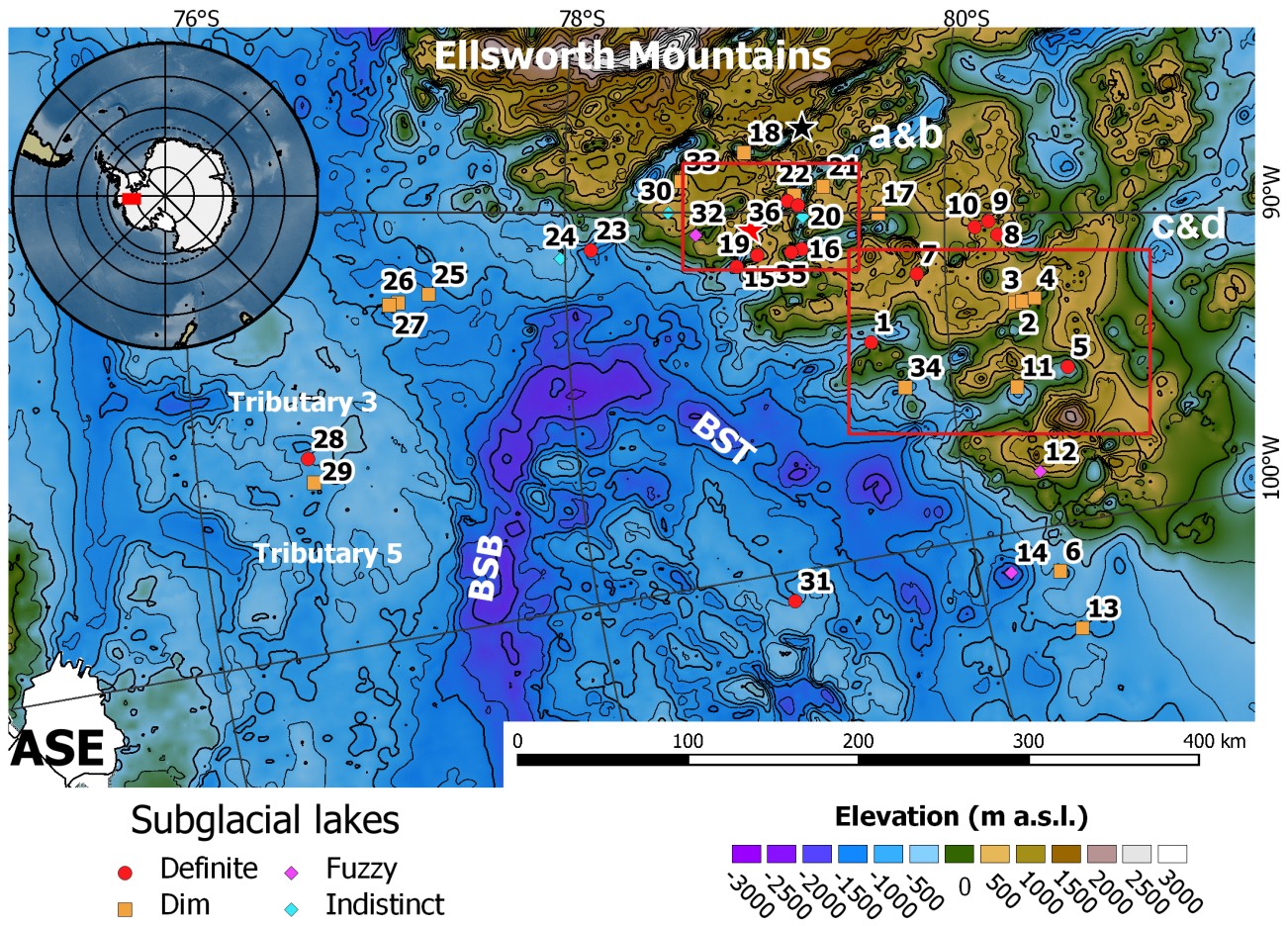

**Figure 6.** Location of subglacial lakes within the Ellsworth Subglacial Highlands. Inset: Location of ESH in Antarctica. The subglacial lakes are represented by different shapes and colours according to the classification of their BRPr. The subglacial lakes are classified from a greater to a lesser degree of confidence, as follow: red dots for definite; yellow squares for dim; pink diamonds for fuzzy; and cyan diamonds for indistinct subglacial lakes. The black and red stars represent SLC and SLE, respectively. In the background the new subglacial DEM and the PIG tributaries are numbered according to the scheme given by Stenoien and Bentley (2000). Contours every 500 m. The red rectangle shows the location of panel a and b (also c and d) in Figure 7. ASE: Amundsen Sea Embayment; BST: Bentley Subglacial Trench; BSB: Byrd Subglacial Basin. Projection: Antarctic Polar Stereographic (EPSG 3031).

pographic contexts. Many exist in linear subglacial troughs with steep side walls. These are a common feature within the ESH and likely reflects a topography which has evolved under conditions of alpine erosion and, before glacial inception, fluvial erosion (Jamieson et al., 2014; Sugden et al., 2017; Vaughan et al., 2007) steered by tectonic influences. Other lakes are located within terrain with reduced relief and a mean elevation below sea level in an area constrained by subglacial hills (Figure 6). This area may have been subject to a mix of areal scour and selective linear erosion beneath the WAIS (e.g., Jamieson et al.,





**Figure 7.** Two main subglacial lakes clusters: Ellsworth Trough and western ESH. The location of this cluster is delimited by red squares in Figure 6. a) First cluster of subglacial lakes in Ellsworth Trough. b) second cluster of subglacial lakes identified in upstream Institute area. The new subglacial DEM is shown in the background. In a) and b) white lines are airborne RES survey tracks. Elevation contours lines at 500 m intervals. c) and d) shows the subglacial water hydrological network, for the same areas of a) and b), plotted in different colours depending the flow direction. The red lines show modelled subglacial drainage towards Amundsen Sea Embayment; the yellow lines, drainage to the Weddell Sea Embayment. Projection: Antarctic Polar Stereographic (EPSG 3031).

2014; Sugden et al., 2017; Paxman et al., 2019).

Subglacial lakes located in the Ellsworth Trough (Ross et al., 2014), in a parallel trough to the west and in adjacent subglacial valleys, lie beneath very slow flowing ice near the ice divide (Figure 9). In contrast, those located in lower topography are distributed along the head of TG, on tributaries 3 and 5 of PIG (SL 28 and 29 in Figure 9), in some topographic depressions

beneath the PIG, and at the northern end of the Ellsworth Mountains beneath the RIS where there is currently a slightly higher





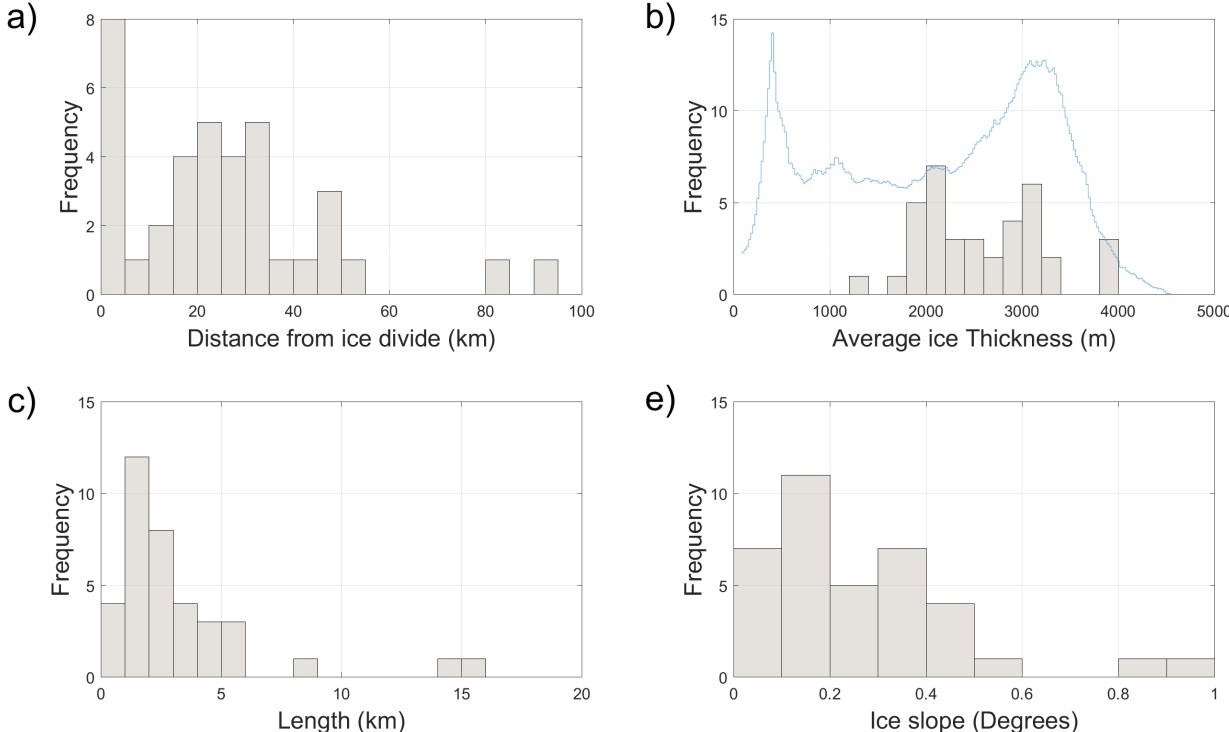

**Figure 8.** Frequency-distribution histograms of subglacial lakes identified in this study. a) Distance from major ice divides. b) Average ice thickness, obtained from BBAS ice thickness measurements.The blue line shows the regional distribution of the ice thickness in Antarctica (Fretwell et al., 2013) for a continental reference. c) Minimum length of identified subglacial lake, calculated by measuring the horizontal extent of the lake reflection. d) Ice surface slope, obtained using Cryosat-2 (Slater et al., 2018).

annual average velocity than elsewhere in the ESH. In a small number of cases (∼10%) however, some subglacial lakes located closer to the PIG outlet over the subglacial topographical barrier do have average overlying velocities close to 60 m/yr (Figure 9). Most of the subglacial lakes (26) are located within the subglacial trough system of the ESH. These subglacial lakes are variable in size but most of them (13) are larger than 2 km. This region also hosts SLC and SLE. In contrast, 11 of the total

identified subglacial lakes are situated along the lowland terrain (below sea level) and are more uniform and larger in size (Figure 8c). Most of these subglacial lakes (21) are greater than 2 km in length and two (26 and 25) relatively large lakes (∼ 15 km) are found in the Bentley Subglacial Highlands towards RIS (Figure 6). Analysis of the ice surface indicates mostly a flat area over the subglacial lakes and a maximum value < 1 degree (Figure 8d) above any of the subglacial lakes.

Based on the assumptions about the shape of subglacial lakes (circular/elliptical) and extrapolation of the observed side-slope

topography at the margin of the subglacial lakes, we estimate a total volume for circular shapes of ∼31km$^3$ (Supplementary



**Figure 9.** Mean annual ice surface velocity (Mouginot et al., 2019) of Pine Island Glacier, Rutford Ice Stream, Institute Ice Stream, Bindschedler Ice Stream and Thwaites Glacier. Black line shows ice surface velocities higher than 250 myr$^{-1}$. The red line indicates the boundary of the water catchment. The blue lines show the subglacial water drainage and the arrows indicates the general flow direction. The Cryosat-2 Elevation model (1 km), virtual hillshade (Helm et al., 2014) is showed in the background. ASE: Amundsen Sea Embayment; BSB: Byrd Subglacial Basin; BST: Bentley Subglacial Trench. Projection: Antarctic Polar Stereographic (EPSG 3031). B) shows the histogram of surface ice velocity over the central part of each subglacial lakes.

information: Table 2) and either $\sim$125 km$^3$ or $\sim$8km$^3$ for elliptical shapes (depending on the use of radius a or b, respectively) stored in all the subglacial lakes, excluding SLE ($\sim$1.4 km$^3$, Woodward et al. (2010)) and SLC ($\sim$2.5 km$^3$, Smith et al. (2019)).





### 3.3 Subglacial flow routing

The water flow routing in the hydrological network initiates near the major ice divides in the region (i.e., RIS-IIS-PIG) and flows
to the margin of the continent (Figure 6d and e). The hydraulic model assumes a wet bed throughout the ESH and the geometry
of the subglacial hydrological network is classically dendritic. It extends almost from the major ice divide (RIS-IIS-PIG) to
the continental margin, connecting the majority (70%) of the subglacial lakes identified in this work into a wider subglacial
hydrological network. We identify 3 main drainage systems: Weddell Sea, Ross Sea and ASE. There are 18 subglacial lakes
draining towards the Weddell Sea; one subglacial lake (dim) draining towards the Ross Sea; and 15 draining towards the
Amundsen Sea. In ASE, most of the subglacial lakes are connected into a single drainage network flowing underneath PIG
and TG, and partially beneath RIS. Only two subglacial lakes are disconnected from the main ASE drainage system: they
originate in the Bentley Subglacial Highlands and drain completely underneath PIG. Significantly, we note that the surface
flow patterns of ice and the flow patterns of subglacial water are not always co-incident. The most evident difference between
hydrological and ice flow catchments are observed underneath the PIG-TG catchments (Figure 7c and 7d). Here, the ice flows
from the PIG-IIS ice divide into the PIG catchment, however the subglacial drainage catchment flows first into the head of the
PIG catchment and then diverts into the TG catchment. We find that the area of the subglacial hydrological catchments of PIG
and TG are $\sim$1.35 x$10^5$ km$^2$ and $\sim$2.83 x$10^5$ km$^2$ respectively (and if calculated using BedMachine are $\sim$1.25 x$10^5$ km$^2$ and
$\sim$2.88 x$10^5$ km$^2$ respectively) and that the ice surface catchment of PIG and TG have areas of $\sim$1.76 x$10^5$ and $\sim$1.86 x$10^5$
respectively (Mouginot et al., 2017).

We find that the majority of the known subglacial lakes in the region coincide with channels delineated in our subglacial
hydraulic modelling. This gives additional confidence to the assessment of the lakes because it provides an indication that
there would be upstream areas which might capture enough water to fill a lake, but also that the channelization would be
appropriately directed to deliver that water into the lake positions. A challenge with the subglacial flow routing is that the bed
DEM is based on sparse data due to the reconnaissance nature of the primary airborne RES data-sets across the study area. As a
consequence, the spline interpolation and the use of potential field data to fill gaps in the DEM means that the routing is subject
to uncertainty associated with overly smooth data in some areas, and potentially noisy data in others. The latter issue would
have most influence in terms of potentially enabling streams to be diverted, but because we use tension spline interpolation we
believe that noise is minimised.

## 4   Discussion

### 4.1   Subglacial lakes and the production of basal water

We have identified several bright basal reflections, which we interpret as West Antarctic subglacial lakes. Many of these are
located at the head of the catchments of PIG and Thwaites Glacier. Many of these subglacial lakes are found in the ESH very
close to the ice divide between IIS, PIG and RIS (Figure 6). Because many of these were classified as 'definite' lakes, we
have confidence that they are likely to be deep water lakes like Lake CECs and Lake Ellsworth. The subglacial water is likely



produced by a combination of basal melting directly over subglacial lakes and from the input of water produced elsewhere in the subglacial catchment.

The spatial distribution of subglacial lakes in terms of size is variable. The RES data shows that 36% of the subglacial lakes are smaller than 2 km and only 8% are bigger than 10 km. Most of the subglacial lakes (56%) are between 2 km and 10 km

in length and are distributed within a region of steep subglacial topographic relief (Ellsworth Subglacial Highlands); while the smaller subglacial lakes are found in much lower subglacial terrain. However, the biggest new subglacial lakes are also located in a low subglacial region, underneath RIS. The distance from the lakes already inventoried to the ice divide is similar to our findings, confirming the tendency of subglacial lakes to be located close to the thick, flat ice and rough basal topography associated with ice divides (Dowdeswell and Siegert, 1999). The ice thickness also has a similar distribution to that described

by Wright and Siegert (2011), very close to the average ice thickness in Antarctica, and most of the subglacial lakes are found beneath an ice thickness around 3000 m. These reported subglacial lakes have the same shape as those identified by Wright and Siegert (2011) but the range of length notably smaller: while Wright and Siegert (2011) mapped subglacial lakes as large as 50 km, the subglacial lakes identified in this work are all less than 20 km in length. Despite this, the predominance of lake length < 10km is similar to the distribution noted by Wright and Siegert (2011).


Subglacial lakes are likely to form beneath areas of thick ice (i.e. > 2.5 km) where the pressure-melting temperature is enough to maintain liquid water (Dowdeswell and Siegert, 1999; Gorman and Siegert, 1999), in areas of internal ice deformation and sliding that contribute heat to produce melting (Siegert et al., 1996), and areas of elevated geothermal heatflux (GHF). Although observations of GHF are absent in the ESH, and modelling efforts (e.g., Shapiro and Ritzwoller, 2004; Maule et al.,

2005; An et al., 2015; Martos et al., 2017) poorly constrained, there are possibilities for elevated GHF, and therefore enhanced production of basal water, in the ESH. One potential localised source of elevated basal heat is enhanced radiogenic heat flux from granite intrusions known to exist within the EWM block (Burton-Johnson et al., 2017; Leat et al., 2018). In addition, the high relief basal morphology of ESH (e.g., Vaughan et al., 2007; Ross et al., 2014; Winter et al., 2015) with its narrow and deep subglacial troughs, will enable a localised intensification of GHF via topographic focusing (van der Veen et al., 2007).

Continent-wide models of the basal thermal regime (e.g., Pattyn, 2010) suggest that the ESH are warm-based throughout, although given the thin ice located between the deep subglacial troughs, it is more likely to have a patchwork of basal thermal regime, with warm-based ice within the deep troughs and cold-based ice on the subglacial interfluves. Beyond the ESH (e.g. PIG, RIS, Thwaites) it is likely that the bed is predominantly warm-based in line with continent-scale models (e.g., Pattyn, 2010). The discrepancy between the Pattyn model and actual basal conditions in the ESH is likely to be due to the coarse

resolution of Bedmap (Lythe and Vaughan, 2001) used by Pattyn (2010). Future numerical modelling of basal thermal regime using our new high-resolution DEM and newly identified subglacial lakes would therefore be an important aspect of improving assessments of the basal thermal regime in this region.





## 4.2 Large-scale subglacial drainage network and lake connectivity

In the ESH, the subglacial water network is mainly controlled by subglacial topography and pre-existing troughs and deep valleys (e.g., Ellsworth and CECs trough) (Siegert et al., 2012). Flow routing into Bentley Subglacial Trench, RIS, IIS and KIS is determined by the combination of the very steep-sided trough walls and the overall form of the ice sheet surface, which partially aligns with ice flow, and drives subglacial water flow along the axis of the deep troughs (Mouginot et al., 2019).

It is likely that the subglacial lakes identified in the Ellsworth trough (Figure 7c) are connected in a very well-defined local drainage system (Siegert et al., 2012; Ross and Siegert, 2019). Modelling of modern hydropotential shows that diversion of subglacial water within this trough (Figure 7c and 7d) is upstream from SLE and most of the seven subglacial lakes in this system are connected or very close to a subglacial water path. It is possible that some episodic events could link these subglacial lakes forming a cascade-type system from high hydraulic areas (i.e. modern ice divide between PIG and IIS) to lower hydraulic

areas (i.e. the Bentley Subglacial Trench). The identification of any such episodic draining (e.g., Wingham et al., 2006) would require analysis of ice elevation changes to capture water infilling/drainage through time. We also identify a dim subglacial lake in the CECs trough, down-ice of SLC toward the RIS (Figure 7c and 7d). Although the current hydraulic modelling does not show a clear connection between this new lake and SLC, it is possible that under different ice sheet configurations both subglacial lakes were connected hydrologically.


## 4.3 The subglacial hydrological catchments of Pine Island and Thwaites Glaciers

We observe that most of the subglacial water draining towards ASE is routed through the Bentley Subglacial Trench in the upper part of the hydrological catchment and driven through the Byrd Subglacial Basin towards the trunk of Thwaites Glacier. The high topography in the mid PIG catchment (Vaughan et al., 2006) means that the hydrological drainage system does not

link to the faster flowing trunk of PIG. Instead, the basal hydrological system is captured by Thwaites. This drainage pattern has two main implications. Firstly, the subglacial hydrological catchments of PIG and Thwaites do not correspond to the ice catchments; they do not coincide either in position or size. Secondly, the hydrological system of TG trunk (Schroeder et al., 2013) may be fed by water sourced in the upper glaciological catchment of PIG, within the ESH.

Any change in the water catchment of the TG, at the head of PIG, could therefore have important glaciological consequences

for the ice dynamics of Thwaites Glacier and the wider ASE. This is particularly critical since the subglacial water drainage area of TG is bigger than previously thought and recent investigations (e.g., Smith et al., 2017) have demonstrated the presence of active subglacial lakes, in a cascade system-type, beneath the trunk of TG. Any water accumulation/drainage (e.g., chain of active subglacial lakes) in this area may affect the basal friction of the ice and therefore the ice flow velocity. Conversely, this pattern may have a reduced importance for PIG in terms of magnitude or timing due to the topographic barrier disconnecting

the drainage upstream with the lower/marginal section of PIG. If we are to clearly understand the potential role of subglacial water on the ice dynamics of the PIG and Thwaites systems, then more investigations of the detailed subglacial and hydrolog-



ical conditions are required.

## 4.4 The spatial relationship between subglacial lakes and ice flow in West Antarctica

Almost 90% of the newly identified subglacial lakes in West Antarctica are located in areas of slow ice flow velocity (<20 m/yr) (Mouginot et al., 2019). There are two likely reasons for this: Firstly, the ice surface slope, which is a crucial driver of basal hydraulic conditions, is typically low in these regions (Supplementary information: Table 1), thus enabling ponding to occur even in relatively low magnitude topographic depressions. Secondly, given the ice flow is slow in this area, we can infer that the subglacial network may be an efficiently draining system that does not enable the pressurization of a deforming bed,
but instead may allow efficient water transport.

## 4.5 Limitations of the RES data, the bed DEM and hydropotential modelling

The BBAS radar data were processed as un-focused SAR images and were based on 1-D reflection (Heliere et al., 2007). In some areas (e.g., subglacial troughs), this provided a poor constraint on the subglacial bed because of the presence of diffracted
hyperbolae caused by an unfocused return of the energy.This means the ice-bed interface is not clear enough to identify in some of the radar lines, and it is especially difficult to detect the bed topography in areas where the strength of the radar return is low. This way of processing could have influenced the BRP calculation, misleading some subglacial lake classification, or underestimating the size of the recognized subglacial lakes due to an uncertainty on which axis (longitudinal or transversal) had been identified.


The BRP calculation was focused in single portions with bright reflectors on each radar profile; and we assume a constant ice attenuation rate by considering the attenuation within these portions as proportional to the ice thickness. Although this attenuation is proportional to the dielectric attenuation, and therefore to the ice temperature and to solute content in the ice (Gudmandsen, 1971; Corr et al., 1993), we did not include any temperature model, which resulted in limited capacity to dis-
tinguish differences in reflectance.

We categorized every subglacial lake based on their physical characteristics using the classification method proposed by Carter et al. (2007) as a guideline. However, some criteria were modified using higher thresholds (e.g., BRP and $\sigma$BRP) as proposed by previous studies (Dowdeswell and Siegert, 2003).


Our new subglacial DEM improves our previous knowledge of subglacial bed condition in the Ellsworth Subglacial Highlands (Bedmap2 (Fretwell et al., 2013), and Bedmachine Antarctica (Morlighem et al., 2019)) and, despite the relatively coarser resolution, it does account for topographic features which connect the Weddell Sea with the Amundsen Sea that are not yet present in other models. We note that our hydropotential model was derived from this new gridded DEM at a resolution of

2 km, and therefore it may not detect some smaller (< 2km) subglacial flow pathways. Although new and/or more detailed
subglacial water or drainage systems could be identified in future RES campaigns, the main drainage pattern would not be
substantially different to that which we have identified under the modern ice sheet configuration.

Additional targeted surveys across the newly identified lake reflections would better constrain the size and area of the sub-
glacial lakes and therefore improve our estimations of the potential volume of subglacial lakes within this region of WAIS.
Moreover, the BBAS survey collected airborne RES using a regular 30-km grid flown at constant elevation (Vaughan and oth-
ers, 2006), which could have missed some features and/or subglacial lakes hosted in areas between each flight line. Therefore,
the number of subglacial lakes discovered in this work could increase with more RES data surveyed using an appropriate flight
geometry.


## 5   Conclusions

1. We used RES data from the 2004/5 BBAS survey to locate subglacial lakes within the EWM region of West Antarc-
   tica. This analysis allowed us to identify 37 new subglacial lakes to add to the existing inventory (Wright and Siegert,
   2012). Assuming a circular shape, we estimate a total subglacial water volume of ~31 km$^3$ (excluding Subglacial Lake
Ellsworth and Subglacial Lake CECs, which are ~1.8 km$^3$ and ~2.9 km$^3$, respectively). While assuming an elliptical
   shape, we obtained a volume of ~124 km$^3$ and ~7.7 km$^3$. We then classified these subglacial lakes according to how
   confident we are in their detection. Using this classification, we identify 13 subglacial lakes with a very high degree of
   confidence. A further 18 'dim' subglacial lakes were also identified.

2. We observed that the majority (75%) of the lakes are situated underneath or close (< 40 km) to the modern ice divide
between Institute Ice Stream and Rutford Ice Stream, Pine Island Glacier and Thwaites Glacier. Furthermore, we also
   detected that slow ice flow is associated with these lakes and that there are always low gradient ice surfaces above them.

3. We developed a new bed DEM based on recently collected survey data that was not previously incorporated in Antarctic
   topographic models. This allowed us to recognize new topographic features such as the long and linear subglacial trough
   systems which connect to multiple sub-catchments and which therefore may play an important role in the basal hydrology
and dynamics of the West Antarctic Ice Sheet.

4. Using the new DEM and the up to date surface elevation model from Cryosat-2 (Slater et al., 2018) we analysed the
   subglacial hydraulic network. We identified the potential for connection between the subglacial lakes and the wider
   subglacial hydrological system, thus providing a mechanism for cascading and active lake drainage. Most importantly,
   however, we show that the hydrological catchments of RIS, PIG and TG do not correspond precisely with glaciological
catchments. Indeed, TG's hydrological catchment appears larger than previously thought, capturing basal water from the
   upper region of PIG.



5. The detection of subglacial lakes was carried out by means of RES, which is based on 1-D reflections. These 1-D reflections do not fully capture the whole body of the subglacial lakes, but we assumed the length presented as a minimum value for each subglacial water body. Consequently, additional targeted surveys across the newly identified lake reflections would better constrain the size of the subglacial lakes.

*Author contributions.* The study was conceived by FN, MB, SJ, NR and AS. BBAS RES data was originally collected and provided by DV, NR, MS. CECs RES data was collected and provided by AR, RZ and JAU. RES processing was undertaken by FN, NR, GG and JAU. DEM was created by FN and GP. RES analysis was undertaken by FN, NR, SJ and MB. The manuscript was written by FN with input from all authors.

*Competing interests.* The authors declare that they have no conflict of interest.

*Acknowledgements.* We acknowledge the British Antarctic Survey (BAS) and Centro de Estudios Científicos (CECs) for providing their radar data for analysis. This work is supported by Natural Environment Research Council (NERC) UK grant NE/J008176/1. FN acknowledges support from the Agencia Nacional de Investigación y Desarrollo (ANID) Programa Becas de Doctorado en el Extranjero, Beca Chile for the doctoral scholarship. The subglacial lakes information is available in supplementary information or from corresponding author.



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
