# Peer review of "Subglacial lakes and hydrology across the Ellsworth Subglacial Highlands, West Antarctica"

_The Cryosphere, 2020_

## Referee Comment (RC1) · Lucas Beem (Referee) · 9 Apr 2020

This article makes the contribution of expanding the inventory of subglacial lakes, specifically in West Antarctica, and discusses the basic characteristics of these lakes. Through the use of radar echo sounding observations that were collected in 2004/2005 bed echoes that could be characterized as a reflection off of an ice/water interface were identified. These reflections were further classified in four categories that relate to the confidence that the echo is from an ice/water interface. The analysis used established approaches for the identification and classification of subglacial water bodies, namely bright, low variance reflections that have flat hydraulic potential.

[Figure]

Overall I found this to be a fine article. I believe the methodology needs clarification in a multiple locations. There are many specific instances that I note within the line comments below. In particular the specularity methodology has left me scratching my head (see line comments 81,122 below). Also, I don't think specularity is being measured/observed, but instead consistency of bed echo is being used as a proxy, in combination with other observables (relative and absolute brightness), to infer specularity. I think that distinction should be clear in the methodology.

I am unmoved by the lake volume estimates. This attempt is highly unconstrained to the point of being untrustworthy. I understand the desire for volume balances beneath the ice sheet and that others have made similar assumptions within the literature, but a volume range from 8 to 125 kmˆ3 with fully contrived shorelines isn't a rigorous or defensible result.

The ice catchment boundaries made it to the abstract, so it must be considered an important detail, but the boundaries (ice verse subglacial hydrology nor old subglacial hydrology verse updated subglacial hydrology) are never directly compared. A figure should be modified to allow the reader to understand this reported observation. The change in 'known' subglacial hydrology catchment beneath Thwaites Glacier may be an important contribution, but the reader has no opportunity to assess this finding and to evaluate the implications to an important region of West Antarctica. Including these boundaries would support the conclusions of the authors and, if truly significant, increase the impact of this article.

When I read though an article I note places where I get confused. Certainly, not every reader would share my confusion, but perhaps some readers would. There are many suggestions below that ask for clarity and specificity. There are also comments that relate to the broader points outlined above.

Line 7) 'region': Not sure a region has been defined yet. 60% increase for Ellsworth Highlands or West Antarctica? While it certainly becomes evident later, the abstract

should stand alone.

23) 'most likely': Hypothesized, maybe?

27)'many that': should it be 'that many' or 'many'?

27,36) 'highly': What is the threshold between dynamic and highly dynamic?

36) Maybe high dynamism *is* the filling and draining, not highly dynamic *and* fills and drains?

39-40) This is at the edge of my grammatical confidence, but are these semi-colons correct?

47) 'Not fully understood': Is anything? Perhaps; "many hypothesis remain untested" or something else

48) 'fastest changing': In what regard? Certainly if we choose differing metrics we could identify different glaciers that are fastest changing. Maybe generalize this statement. If PIG and THW are the fastest changing then e.g. should be i.e.

52) 'ice basal water': perhaps just 'basal water'?

52-53) 'edge of the continent': This might not be accurate as the ice sheet grounding zone is not coincident with the continental margin.

56) First time PIG is used and acronym is not defined.

62) 'subset of this data-set': Which subset? Just the data over ESH? Why was this done? The justification for this work talks about the importance of analyzing existing data sets? Why wasn't the entire data set analyzed / why restrict it?

65) How parenthetical abbreviations with parenthetical references are formatted varies. For example: Line 65 they are within the same parenthetical. Line 39-40 they are in independent parenthetical. Should this be consistent throughout, or maybe rewrite to not require the combination of parentheticals?

66-69) Maybe explicitly enumerate the list, e.g. i) ii), for clarity

73-75) I think the sentence would read more clearly if the final clause was moved closer to the beginning of the sentence, maybe 'Following previously published methods, these reflections were then...'

78) Point 2 is that the lake surface is has a flat hydraulic potential, right? Should it use those terms? The hydraulic potential water surface should not be inclined in a lake context. I wonder if 'potential', means a candidate site and not a potential energy surface, if so, that is leading to my confusion. Also it's unclear what 11 times opposite means. Perhaps it's, 11 times the magnitude in the opposite direction?

79) What's distinguishes between points 2 and 3? Appears to be describing the same characteristic in differing terms, flat hydraulic potential.

81) First time BRPr is used and it is not defined. If BPRr is a proxy for specularity, the term BPRr is not used in section 2.3. On line 121, BPRr appears to be absolute reflectivity.

82) '<3sigma BRPr': See line 122 comment

88) Is BRPe attenuation in ice or energy loss more generally? In this line, BRPe is defined by the methodology used to quantify it.

90,97) Is there an inconsistency in variables used for geometrical spreading ($L_t$ and $L_g$)? $L_t$ never appears in an equation. Also, with the placement of the parenthetical $L_i$, in line 90, it suggests it is defined differently than in line 97. Is $L_i$ attenuation, system gains, or attenuation and system gains? I would look to make this nomenclature consistent and unambiguous. Also, the terms are repeatedly defined, maybe simplify to reduce repetition. An Oxford comma on line 90 could serve to reduce ambiguity. Check for consistency in the use of Oxford commas. Appears in some lists, for instance sentence that begins line 304 and not in others, for instance line 90.

99) Is an indent missing?

102) 'height': Is that height of the antenna above the ice surface?

104) Is an indent missing?

104) 'section': What is a 'section'? How long/how many samples? This should be clear how calculated attenuation values vary within the survey and over what length scale. Understanding attenuation application could be significant to using BPR as an identifier for bright bed. On line 376, attenuation rate is reported as 'constant'. But here on 104, its seems to be calculated on a 'section' by 'section' basis. Which is it? Also, reporting the magnitude of attenuation rate will be of interest to the community.

111) 40 samples on either side or 20 samples are either side. What is a nominal sample spacing? If statistics are calculated for 40 meters of bed verses or hundreds of meters or kilometers it will influence significance/usability of the results.

Section 2.4) Which of the categories require hydrologic-potential flatness? Only 'definite' explicitly includes hydrologic-potential. Can a sloped hydraulic-potential surface be considered a lake with the other classifications?

122) '<3sigma BPRr': Maybe the threshold magnitude of specularity proxy should be defined here. It is unclear to me. Partially as I am confused about the distinction between BRP and BRPr (see line 81 comment) and what <3sigma BRPr means. I understand that the analysis requires a low magnitude of standard deviation to be a definite lake, but that magnitude is not defined. Does the threshold vary between lakes, or is a universal threshold applied? Looking at Carter et al., 2007 those authors used 3 db standard deviation in bed echo strength as a threshold for a specularity proxy. Should '<3sigma BRPr' be a 1sigma threshold of less than 3 db?

123) 'flat hydraulic surface': flat hydraulic potential surface

133) 'distinguish': What characteristics are not distinct? Should the clause be more specific?

Section 2.5) Lake shape assumptions seem poorly justified. Why should lakes have the

same aspect ratio? The average of two lakes (SLE, SLC) does not revel much about a distribution of lake sizes. How do aspect ratios of pater-noster lakes vary within a sub-aerial valley, does this lend credence to this assumption? Not all the candidate lakes are within a trough. Are the trough assumptions applied to all environmental settings? If so, how can that be justified? Perhaps making volume estimates from a single RES crossing exceeds the capacities of the data.

147) A mention of the tectonic environment (like the details discussed near line 235) in these section would support the choice of a side slope lake depth assumption.

156) 'replaced them with Bedmap2': Bedmap2 is 1km grid product. Which bedmap2 value was chosen for inclusion in the new 2km grid DEM? What methodology was employed

158) 'downsampled': How?

162-163) Why include units for some variables?

167) equation 5, g is a different typeface.

171) These citations are specific to the middle of Whillans ice stream. The lakes in this article are in a different glaciological setting. How does hydrology in fine grained subglacial substrate in the middle of a fast flowing ice stream relate to the hydraulics in a fault bounded subglacial highland trough beneath an ice divide?

177) 'tends': Does it ever not?

218) 'very close' is greater than 20 km? What does 'very close' mean?

220) '17': 17 is the total number of 'small lakes' or the number of small lakes near the divide? Maybe if it said (17 of x) or (x of 17) whichever is correct. Would that be clearer if the numeric values of this section where not parenthetical but part of the sentence, e.g. 'Seventeen of the small lakes. . .',?

221) 'these': Which lakes are 'these'? only the 3 largest?

228) 'mean': What do we learn from the mean? Would the mode be more descriptive?

241) Where is the ET? Geographical names should be locatable with labels on figures. Particularly with a reference to figure 9 which does not have any locatable basal topographic features.

246) All the others have a count, why use percentage here? Is it better to be consistent?

246-253) Seems like some of this is repetitious. Velocity description occurs on line 228, lowland description occurs on line 238. Length appears on line 227.

262) Percentage or count? consistency?

269) '(Figure 7c and 7d)': These panels do not show catchment boundaries, so it is not possible to detect how the subglacial hydrology catchments and ice catchments differ and how that might be an important insight.

275) Is an indent missing?

277) 'channelization': Is channelization an assumption? How is the geometry of the system known? Perhaps 'routing' is a better word?

289) How deep is 'deep'? 290) What is the evidence of melting over the lake? Perhaps present as a hypothesis?

293) What is a 'variable' distribution? Can a more specific statement be used?

301) How is the shape of these lakes known? They are assumed to be circular or elliptical. How can these shapes be compared to the shapes in the Wright and Siegert inventory? In Wright and Siegert inventory a single length value is reported except for 8 lakes which have an additional width value. How is any meaningful shape comparison accomplished with these data?

306-308) It is ambiguous if this statement is an inclusive list (all are necessary) or are

three independent criteria. I might rewrite to have the distinction be explicit.

330) 'trough': Capitalize?

334)'cascade-type system': This term is used a few times (line 352,418) without a clear definition of what characterizes this system or what other systems might exist. I presume 'cascade' refers to a temporal correlation between respective draining and filling events? My understanding only becomes a possibility after reference to Thwaites lakes from Smith et al. as cascade. Maybe clearer terminology is needed?

349) Is an indent missing? Section 4.5) Much of this section appears to be methodology to me. Consider moving the text.

370) No space after 'energy.'

376) 'focused'/'single portions': Isn't BRP calculated everywhere? What does 'single portions' mean? Is it a length of flight line, or a certain number of samples? If so, that should be explicitly stated with the magnitude (e..g. # samples) of data used.

396) 'elevation': should it be 'altitude'?

398) 'appropriate': What is 'appropriate'? Denser (more closely spaced) survey lines are needed?

406)'124' and '7.7': Different magnitudes than reported on line 257

408) ''dim'': Dim in quotes here, but not elsewhere. Which way should it be?

FIGURES Figure 1) Colorbar: 3000 is white. But back ground is white as are the masks for ice shelfs. Maybe change the end member color or background.

Figure 1) Colorbar: Mapping of elevation order with negative elevations closer to top of page is counter to more intuitive mappings of high elevation above lower elevation.

Figure 1) Figure 1 should include all the places referenced in the text. Should all abbreviations used in the figure be defined in the caption? This might assist the reader

in establishing geographic spatial relationships.

Figure 1) Adding the flight lines to Figure 1 would aid in establishing the geographical extent of the aerial survey. Figure 4 is the only depiction of the survey extent and is plotted over only bed elevation with a minimum of geographic place names. I think having the flight lines on an image with more geographic and physical detail will help orient the reader.

Figure 6) 'a&b' and 'c&d',: maybe include reference to Figure 7.

Figure 8) Caption 'regional distribution in Antarctica': What does that mean? Is this the ice thickness distribution for the BBAS survey, all of Antarctica?

Figure 9) Why group a histogram of surface velocities with a map of hydrology routing? Figure 9b, should be in Figure 8.

SUPPLEMENT Table 2) The use of both BRP (in caption) and BRPr (table header) without a clear definition of difference.
* * *

---

## Referee Comment (RC2) · Anonymous Referee #2 · 23 Apr 2020

This paper identifies 37 new subglacial lakes in West Antarctica from ice-penetrating radar data. Radiometric properties were used to classify the confidence of these lakes. A volume estimate was made for these lakes. New topography measurements were used to make an updated DEM of the Ellsworth region so that a water routing model could be generated to investigate the potential for drainage.

This work is an important contribution to lake inventories and hydrological understanding, though some areas of this paper require clarification or further discussion. The volume estimates do not seem particularly meaningful given the assumptions made in the methods and the uncertainty of the results (see comments below). Given that the

water routing model is the primary evidence for connected drainage, it would be useful to include more information on the topography data (e.g. survey spacing, data density). Also, improved topography is an important contribution, and the impact could be enhanced by providing quantitative information on the improvement or showing comparisons to Bedmap2.

There are some statements that seem to conflate active and stable lakes (see comments on lines 37-40), and I believe there could be more discussion on which category the newly discovered lakes fall into. Generally speaking, active lakes identified with satellite observations do not have a clear radar signature, and RES-detected lakes are not observed to have surface elevation changes. The authors hypothesize that these lakes are part of a dynamic drainage system and speculate about cascade-type drainage. It is fine to suggest this, but the fact that many of the lakes in this study are "definite" RES-detected lakes indicates that they could very well fall into the inactive RES lake category. So far, no active drainage has been observed in this region. Previous investigations of SLC and SLE have concluded that these lakes are stable. Perhaps there is a more nuanced stance where RES lakes can be part of a drainage system without the dramatic ice surface drop of active lakes, and the authors do acknowledge that satellite observations of change would be required to confirm drainage. But I think it is important that the authors discuss these contradictory pieces of evidence.

Line 16: "reported acceleration of ice velocity"

Reported an acceleration of ice velocity?

Line 37-40: "These active subglacial lakes have been identified using a range of techniques including satellite measurements of ice surface elevation changes (e.g., Wingham et al., 2006; Smith et al., 2009), characterisation of the subglacial topography from ice surface data (e.g., Bell et al., 2007; Bell, 2008; Jamieson et al., 2016); airborne radio echo sounding (RES) (e.g., Robin et al., 1970; Popov and Masolov, 2003); and/or

ground-based RES (e.g., Rivera et al., 2015)."

It is unclear what is meant by the identification of lakes through the "characterisation of the subglacial topography from ice surface data." Bell et al. (2007) detected active lakes using satellite data, similarly to Wingham et al. (2006) and Smith et al. (2009). Bell (2008) reviews subglacial lakes gathered from a variety of different sources and surveys, including active lakes detected from satellite data, and non-active lakes detected with radar. The Jamieson et al. (2016) study does not identify lakes. Rather, they hypothesize about potential lake locations by running a water routing model on estimated bed topography.

Was this intended to be a statement about active lakes, or subglacial lakes in general? To the best of my knowledge, neither Robin et al. (1970) or Popov and Masolov (2003) have identified active lakes; the lakes they found are considered stable. The Rivera et al. (2015) study also concluded that their lake was stable. The only study that I am aware of that has seen any radiometric evidence for active lakes is Langley et al. (2011):

Langley, K., Kohler, J., Matsuoka, K., Sinisalo, A., Scambos, T., Neumann, T., ... & Albert, M. (2011). Recovery Lakes, East Antarctica: Radar assessment of sub­glacial water extent. Geophysical Research Letters, 38(5).

Line 47-49: "Given the fact that this region is located up-ice of the fastest-changing ice streams in the world (e.g., Pine Island Glacier and Thwaites Glacier), and that they are some of the most vulnerable glaciers to ongoing climate change (Martin et al., 2019)..."

This statement could be better cited. Some options:

Rignot, E., Mouginot, J., Morlighem, M., Seroussi, H., & Scheuchl, B. (2014). Widespread, rapid grounding line retreat of Pine Island, Thwaites, Smith, and Kohler glaciers, West Antarctica, from 1992 to 2011. Geophysical Research Letters, 41(10), 3502-3509.

Rignot, E., Mouginot, J., Scheuchl, B., van den Broeke, M., van Wessem, M. J., & Morlighem, M. (2019). Four decades of Antarctic Ice Sheet mass balance from 1979–2017. Proceedings of the National Academy of Sciences, 116(4), 1095-1103.

Joughin, I., Smith, B. E., & Medley, B. (2014). Marine ice sheet collapse potentially under way for the Thwaites Glacier Basin, West Antarctica. Science, 344(6185), 735-738.

Line 85: The BBAS acronym is not defined

Section 2.3: There are two different definitions of specularity. It might be helpful to explain that the method in this manuscript is different from the specularity calculation method in Schroeder et al. (2014) to avoid confusion.

Line 139: "in a GIS"

In a GIS software?

Section 2.5: The methods for surface area and volume calculation are not convincing. The dimensions of two lakes (SLC and SLE) do not provide a statistically robust or representative basis for the dimensions of other lakes. The lake geometry outlined in Figure 3 seems unrealistic, especially if a lake does not happen to fall within a trough or if there is sediment infill. And since the data is not SAR processed, isn't it possible that the hyperbola slopes are different from the actual topography?

Section 2.6: Given that the water routing model is the primary piece of evidence for the connected drainage hypothesis, I think you can elaborate on the topography. It would be helpful to include information on radar grid spacing, changes from Bedmap2, or percentage of grid cells within 5km of a data point.

Lines 255-256: Is the order of magnitude uncertainty in cumulative lake volume really make this a meaningful result, especially given the assumptions in volume calculation?

Line 302: "the range of length notably smaller"

The range of length is notably smaller?

Line 310: "poorly constrained"

Are poorly constrained?

Section 4.2: Is it being hypothesized that these lakes are active lakes? It should be discussed that radar-detected lakes do not have drainage that can be observed from the surface. Or maybe radar-detected lakes are more dynamic than previously expected, but not active enough to be seen from the surface? If you are hypothesizing a more active regime, it might be helpful to cite the MacKie et al. (2020) study which predicts that there are active lakes in the Ellsworth region.

MacKie, E. J., Schroeder, D. M., Caers, J., Siegfried, M. R., & Scheidt, C. (2020). Antarctic Topographic Realizations and Geostatistical Modeling Used to Map Subglacial Lakes. Journal of Geophysical Research: Earth Surface.

Line 335-335: "The identification of any such episodic draining (e.g., Wingham et al., 2006) would require analysis of ice elevation changes to capture water infilling/drainage through time."

Following on the comment above, there'll have to be a thoughtful statement about why you think these lakes could be detected by satellite, when so far all other radar-detected lakes (with a possible few exceptions) have not.

Line 337: "down-ice"

Downstream?

423-424: Why is it assumed that the observed lake lengths are the minimum length?

---

## Author Comment (AC1) · 10 Jun 2020

**Responses to Lucas Beem for: "Subglacial lakes and hydrology across the Ellsworth Subglacial Highlands, West Antarctica" (MC No: tc-2020-68).**

We very much appreciate the insightful comments from Dr. Lucas Beem (*italic* and highlighted in grey) and for his constructive and helpful reviews of our manuscript. Below we respond (non-highlighted text) to the comments.

**Lucas Beem comments**

*This article makes the contribution of expanding the inventory of subglacial lakes, specifically in West Antarctica, and discusses the basic characteristics of these lakes. Through the use of radar echo sounding observations that were collected in 2004/2005 bed echoes that could be characterized as a reflection off of an ice/water interface were identified. These reflections were further classified in four categories that relate to the confidence that the echo is from an ice/water interface. The analysis used established approaches for the identification and classification of subglacial water bodies, namely bright, low variance reflections that have flat hydraulic potential.*

*Overall I found this to be a fine article. I believe the methodology needs clarification in a multiple locations. There are many specific instances that I note within the line comments below. In particular the specularity methodology has left me scratching my head (see line comments 81,122 below). Also, I don't think specularity is being measured/observed, but instead consistency of bed echo is being used as a proxy, in combination with other observables (relative and absolute brightness), to infer specularity. I think that distinction should be clear in the methodology*

We will alter the text to make it explicit that we use the consistency of the bed echo as a proxy to infer the specularity with reference to key papers (e.g., Peters et al., 2005; Carter et al., 2007).

*I am unmoved by the lake volume estimates. This attempt is highly unconstrained to the point of being untrustworthy. I understand the desire for volume balances beneath the ice sheet and that others have made similar assumptions within the literature, but a volume range from 8 to 125 km^3 with fully contrived shorelines isn't a rigorous or defensible result.*

We agreed. This estimation may be an overinterpretation of the data we have. We will remove the area and volume estimations due to the high uncertainties and will modify the discussion to account for this. It is a small component of the analysis and will not impact the key findings of the paper in terms of the new inventory or lakes.

*The ice catchment boundaries made it to the abstract, so it must be considered an important detail, but the boundaries (ice verse subglacial hydrology nor old subglacial hydrology verse updated subglacial hydrology) are never directly compared. A figure should be modified to allow the reader to understand this reported observation. The change in 'known' subglacial hydrology catchment beneath Thwaites Glacier may be an important contribution, but the reader has no opportunity to assess this finding and to evaluate the implications to an important region of West Antarctica. Including these boundaries would support the conclusions of the authors and, if truly significant, increase the impact of this article.*

We will include more detail in the results and discussion about the subglacial hydrological catchment. In particular we will include a figure with the subglacial hydrological drainage system simulated with Bedmap2 and our new DEM and will provide quantitative differences. Additionally, we will include more discussion on this point.

*When I read though an article I note places where I get confused. Certainly, not every reader would share my confusion, but perhaps some readers would. There are many suggestions below that ask for clarity and specificity. There are also comments that relate to the broader points outlined above.*

We address each comment below and in addition we will work on further proof reading the manuscript to ensure clarity throughout. Our responses to the specific comments are detailed below.

**Specific comments:**

*Line 7) 'region': Not sure a region has been defined yet. 60% increase for Ellsworth Highlands or West Antarctica? While it certainly becomes evident later, the abstract should stand alone.*

We will change to West Antarctica.

*23) 'most likely': Hypothesized, maybe?*

We will change this so it will be clear we are hypothesizing this.

*27)'many that': should it be 'that many' or 'many'?*

We will change to "that many".

*27,36) 'highly': What is the threshold between dynamic and highly dynamic?*

We will delete the word "highly", in both cases.

*36) Maybe high dynamism \*is\* the filling and draining, not highly dynamic \*and\* fills and drains?*

Yes, the word should be "is". We will amend the text.

*39-40) This is at the edge of my grammatical confidence, but are these semi-colons correct?*

We will change the semi-colons on lines 39 and 40 to commas.

*47) 'Not fully understood': Is anything? Perhaps; "many hypothesis remain untested" or something else*

We will alter the text to: 'many hypotheses remain untested'.

*48) 'fastest changing': In what regard? Certainly if we choose differing metrics we could identify different glaciers that are fastest changing. Maybe generalize this statement. If PIG and THW are the fastest changing then e.g. should be i.e.*

We will change this from e.g. to i.e.

*52) 'ice basal water': perhaps just 'basal water'?*

We will be more concise and will change the text as suggested.

*52-53) 'edge of the continent': This might not be accurate as the ice sheet grounding zone is not coincident with the continental margin.*

We will alter the text to "ice sheet grounding zone".

*56) First time PIG is used and acronym is not defined.*

PIG will be defined at first mention.

*62) 'subset of this data-set': Which subset? Just the data over ESH? Why was this done? The justification for this work talks about the importance of analyzing existing data sets? Why wasn't the entire data set analyzed / why restrict it?*

We analysed the available BBAS data from season 2004/2005. We will make it clear in the text.

*65) How parenthetical abbreviations with parenthetical references are formatted varies. For example: Line 65 they are within the same parenthetical. Line 39-40 they are in independent parenthetical. Should this be consistent throughout, or maybe rewrite to not require the combination of parentheticals?*

We will change the referencing to make it consistent throughout the text.

*66-69) Maybe explicitly enumerate the list, e.g. i) ii), for clarity*

We will change to an enumerated list.

*73-75) I think the sentence would read more clearly if the final clause was moved closer to the beginning of the sentence, maybe 'Following previously published methods, these reflections were then...'*

We will change as suggested.

*78) Point 2 is that the lake surface is has a flat hydraulic potential, right? Should it use those terms? The hydraulic potential water surface should not be inclined in a lake context. I wonder if 'potential', means a candidate site and not a potential energy surface, if so, that is leading to my confusion. Also it's unclear what 11 times opposite means. Perhaps it's, 11 times the magnitude in the opposite direction?*

We will rewrite this and make it clearer. We will change the text so that it will be clear that we mean a candidate site and that the 11 times does indeed mean 11 times magnitude in the opposite direction.

*79) What's distinguishes between points 2 and 3? Appears to be describing the same characteristic in differing terms, flat hydraulic potential.*

Point 2 is referred to the ice surface, whereas point 3 is referred to the hydraulic potential. We will clarify this in the text.

*81) First time BRPr is used and it is not defined. If BPRr is a proxy for specularity, the term BPRr is not used in section 2.3. On line 121, BPRr appears to be absolute reflectivity*

It should be read BRP. The BRPr will be removed and replaced by BRP instead.

*82) '<3sigma BRPr': See line 122 comment*

We will clarify this point in the text so that it will be clear that is 3 dB sigma BRP.

*88) Is BRPe attenuation in ice or energy loss more generally? In this line, BRPe is defined by the methodology used to quantify it.*

BRPe refers to energy corrected for both geometrical spreading and ice attenuation. We will make this clear in the text.

*90,97) Is there an inconsistency in variables used for geometrical spreading ($L_t$ and $L_g$)? $L_t$ never appears in an equation. Also, with the placement of the parenthetical $L_i$, in line 90, it suggests it is defined differently than in line 97. Is $L_i$ attenuation, system gains, or attenuation and system gains? I would look to make this nomenclature consistent and unambiguous. Also, the terms are repeatedly defined, maybe simplify to reduce repetition.*

$L_t$ should not be on the equation. We will review this section so that this point will be clear. We will check the parenthesis and all the variables included in the equation. Similarly, we will check the nomenclature is consistent throughout manuscript.

*102) 'height': Is that height of the antenna above the ice surface?*

We will rewrite this in the text so that it will clear that we meant the height of antenna above the ice surface.

*104) Is an indent missing?*

Indent will be added.

*104) 'section': What is a 'section'? How long/how many samples? This should be clear how calculated attenuation values vary within the survey and over what length scale. Understanding attenuation application could be significant to using BPR as an identifier for bright bed. On line 376, attenuation rate is reported as 'constant'. But here on 104, its seems to be calculated on a 'section' by 'section' basis. Which is it? Also, reporting the magnitude of attenuation rate will be of interest to the community.*

The attenuation rate was calculated using variable numbers of sample points depending on the particular area. For example, it ranges from 120 samples in the smallest section (within steep terrain) to more than 500 samples before and after a subglacial lake candidate depending on the basal interface. We will detail this in the supplementary information.

*111) 40 samples on either side or 20 samples are either side. What is a nominal sample spacing? If statistics are calculated for 40 meters of bed verses or hundreds of meters or kilometers it will influence significance/usability of the results*

The standard deviation was calculated 20 samples either side of a point. The horizontal resolution of the radar is ~21 m. We will explicit this in the text.

*Section 2.4) Which of the categories require hydrologic-potential flatness? Only 'definite' explicitly includes hydrologic-potential. Can a sloped hydraulic-potential surface be considered a lake with the other classifications?*

All the categories require a hydropotential surface that is more linear than its surroundings, but the smoother the lake, the more likely it is to be defined as 'definite'. However, we also note that this surface may be tilted if the subglacial lakes are smaller than 4-5 km. As a consequence, the term 'flat' may not be appropriate and instead we suggest we will apply the term 'smooth' or 'linear'. We will clarify this in the text.

*122) '3σ BRPr': Maybe the threshold magnitude of specularity proxy should be defined here. It is unclear to me. Partially as I am confused about the distinction between BRP and BRPr (see line 81 comment) and what '3 BRPr' means. I understand that the analysis requires a low magnitude of standard deviation to be a definite lake, but that magnitude is not defined. Does the threshold vary between lakes, or is a universal threshold applied? Looking at Carter et al., 2007 those authors used 3 db standard deviation in bed echo strength as a threshold for a specularity proxy. Should '<3σ BRPr' be a 1sigma threshold of less than 3 db?*

We used the same threshold for every subglacial lake candidate. This threshold should have been written as 3 dB $\sigma$ BRPe. Thank you for noticing this important detail.

*123) 'flat hydraulic surface': flat hydraulic potential surface*

We will change the text as suggested.

*133) 'distinguish': What characteristics are not distinct? Should the clause be more specific?*

We will change the clause to make it so that it will be clear that we meant the difficulty to recognise each subglacial lake reflectivity from the surrounding bed reflectivity.

*Section 2.5) Lake shape assumptions seem poorly justified. Why should lakes have the same aspect ratio? The average of two lakes (SLE, SLC) does not revel much about a distribution of lake sizes. How do aspect ratios of pater-noster lakes vary within a subaerial valley, does this lend credence to this assumption? Not all the candidate lakes are within a trough. Are the trough assumptions applied to all environmental settings? If so, how can that be justified? Perhaps making volume estimates from a single RES crossing exceeds the capacities of the data.*

We recognise there are a large variation in the estimation of each potential subglacial lakes volume. We will delete these estimations in the manuscript to avoid overinterpretation.

*147) A mention of the tectonic environment (like the details discussed near line 235) in these section would support the choice of a side slope lake depth assumption.*

We will remove the depth estimations of subglacial lakes.

*156) 'replaced them with Bedmap2': Bedmap2 is 1km grid product. Which bedmap2 value was chosen for inclusion in the new 2km grid DEM? What methodology was employed*

We use the nearest pixel to interpolation and create the downsampled output pixel value (nearest neighbor algorithm). We will include this in the text

*158) 'downsampled': How?*

We use the nearest neighbor routine. We will clarify this in the text.

*162-163) Why include units for some variables?*

We will include the units for all the variables and indicate what density we use for water and ice.

*167) equation 5, g is a different typeface*

We will change the typeface.

*171) These citations are specific to the middle of Whillans ice stream. The lakes in this article are in a different glaciological setting. How does hydrology in fine grained subglacial substrate in the middle of a fast flowing ice stream relate to the hydraulics in a fault bounded subglacial highland trough beneath an ice divide?*

We will rewrite this paragraph and indicate that we followed previous investigations in Ellsworth Subglacial Highlands (e.g., Vaughan et al 2007; Rivera et al., 2015) where water pressure was assumed to be close to the overburden ice pressure in subglacial flow path calculations.

*177) 'tends': Does it ever not?*

Yes - it does. We will remove the word tends.

*218) 'very close' is greater than 20 km? What does 'very close' mean?*

We will change "very close" to "less than 50 km".

*220) '17': 17 is the total number of 'small lakes' or the number of small lakes near the divide? Maybe if it said (17 of x) or (x of 17) whichever is correct. Would that be clearer if the numeric values of this section where not parenthetical but part of the sentence, e.g. 'Seventeen of the small lakes. . .',?*

We will change the numeric value of this example and include it in the sentence.

*221) 'these': Which lakes are 'these'? only the 3 largest?*

We referred to the small subglacial lakes. We will change this to "over small subglacial lakes".

*228) 'mean': What do we learn from the mean? Would the mode be more descriptive?*

We will include the mode in the supplementary material. Thanks for the suggestion.

*241) Where is the ET? Geographical names should be locatable with labels on figures. Particularly with a reference to figure 9 which does not have any locatable basal topographic features.*

We will include more geographical names in the figure to provide a clearer spatial reference.

*246) All the others have a count, why use percentage here? Is it better to be consistent?*

We will change it to a count.

*246-253) Seems like some of this is repetitious. Velocity description occurs on line 228, lowland description occurs on line 238. Length appears on line 227.*

We will review this paragraph and we will make it clearer and to avoid repetition.

*262) Percentage or count? consistency?*

We will change this to count.

*269) '(Figure 7c and 7d)': These panels do not show catchment boundaries, so it is not possible to detect how the subglacial hydrology catchments and ice catchments differ and how that might be an important insight.*

We will include the catchment boundaries in suggested figures and will also bolster the text to refer to the figure and describe the way they differ.

*275) Is an indent missing?*

Yes – we will add it.

*277) 'channelization': Is channelization an assumption? How is the geometry of the system known? Perhaps 'routing' is a better word?*

Thank you for this clarifying suggestion. We will change the wording to "routing".

*289) How deep is 'deep'?*

We will clarify this in the text by changing the sentence to "(. . . ) deeper than 100 m as are SLC and SLE".

*290) What is the evidence of melting over the lake? Perhaps present as a hypothesis?*

Thank you for noting this – you are correct that we do not have any measurements so we will therefore introduce this as a hypothesis.

*293) What is a 'variable' distribution? Can a more specific statement be used?*

The distribution of the subglacial lakes does not have an evident pattern to its distribution. We will clarify this in the text.

*301) How is the shape of these lakes known? They are assumed to be circular or elliptical. How can these shapes be compared to the shapes in the Wright and Siegert inventory? In Wright and Siegert inventory a single length value is reported except for 8 lakes which have an additional width value. How is any meaningful shape comparison accomplished with these data?*

We recognize this may be too speculative since we don't have a fully understand of the subglacial water bodies. We will delete these conjectures as it distracts from the main focus of the paper which is to simply identify the lakes and discuss their spatial distribution.

*306-308) It is ambiguous if this statement is an inclusive list (all are necessary) or are three independent criteria. I might rewrite to have the distinction be explicit.*

This is not an inclusive list. These are single criteria suitable for the occurrence of subglacial water. We will rewrite it to make it clearer. Thanks for the suggestion.

*330) 'trough': Capitalize?*

We will capitalize T.

*334)'cascade-type system': This term is used a few times (line 352,418) without a clear definition of what characterizes this system or what other systems might exist. I presume 'cascade' refers to a temporal correlation between respective draining and filling events? My understanding only becomes a possibility after reference to Thwaites lakes from Smith et al. as cascade. Maybe clearer terminology is needed?*

We will clarify this concept at its first mention.

*349) Is an indent missing? Section 4.5) Much of this section appears to be methodology to me. Consider moving the text.*

Indent will be added.

*370) No space after 'energy.'*

Space will be added.

*376) 'focused'/'single portions': Isn't BRP calculated everywhere? What does 'single portions' mean? Is it a length of flight line, or a certain number of samples? If so, that should be explicitly stated with the magnitude (e.g. # samples) of data used.*

The BRP was calculated for each subglacial candidate within a determined number of samples for each subglacial candidate. This number of samples is variable, and it will be added in the Annex table.

*396) 'elevation': should it be 'altitude'?*

We will change this to altitude.

*398) 'appropriate': What is 'appropriate'? Denser (more closely spaced) survey lines are needed?*

We meant an optimized survey to characterize the subglacial interface, considering the geometry of the subglacial topography. In other words, we would ideally have lines going across topographical features directly as opposed to diagonally (e.g., across subglacial trough).

*406)'124' and '7.7': Different magnitudes than reported on line 257*

All these estimations will be deleted.

*408) ''dim": Dim in quotes here, but not elsewhere. Which way should it be?*

Quotes will be deleted.

*Figure 1) Colorbar: 3000 is white. But back ground is white as are the masks for ice shelfs. Maybe change the end member color or background.*

We will change the colour of the ice shelves.

*Figure 1) Colorbar: Mapping of elevation order with negative elevations closer to top of page is counter to more intuitive mappings of high elevation above lower elevation.*

We will change as suggested. Thanks.

*Figure 1) Figure 1 should include all the places referenced in the text. Should all abbreviations used in the figure be defined in the caption? This might assist the reader*

Thanks for this suggestion. We will complement the caption.

*Figure 6) 'a&b' and 'c&d',: maybe include reference to Figure 7.*

We will include the reference to Figure 7.

*Figure 8) Caption 'regional distribution in Antarctica': What does that mean? Is this the ice thickness distribution for the BBAS survey, all of Antarctica?*

This is the regional distribution for all of Antarctica (values taken from Bedmap2).

*Figure 9) Why group a histogram of surface velocities with a map of hydrology routing? Figure 9b, should be in Figure 8.*

We thought it would be more useful for the reader to have this histogram to add context to the hydrology map. The spatial distribution of the velocity are shown in the map of the study area so we did not wish to repeat it.

*Table 2) The use of both BRP (in caption) and BRPr (table header) without a clear definition of difference.*

We have now defined both terms in the text and added a clarification footnote.

---

## Author Comment (AC2) · 10 Jun 2020

**Responses to Reviewer 2 for: "Subglacial lakes and hydrology across the Ellsworth Subglacial Highlands, West Antarctica" (MC No: tc-2020-68).**

We are grateful to reviewer 2 for the helpful reviews of our manuscript and for the references provided. Below we respond (non-highlighted text) to the comments of reviewer 2 (*italic* and highlighted in grey).

**Anonymous referee #2**

*This paper identifies 37 new subglacial lakes in West Antarctica from ice-penetrating radar data. Radiometric properties were used to classify the confidence of these lakes. A volume estimate was made for these lakes. New topography measurements were used to make an updated DEM of the Ellsworth region so that a water routing model could be generated to investigate the potential for drainage. This work is an important contribution to lake inventories and hydrological understanding, though some areas of this paper require clarification or further discussion. The volume estimates do not seem particularly meaningful given the assumptions made in the methods and the uncertainty of the results (see comments below). Given that the water routing model is the primary evidence for connected drainage, it would be useful to include more information on the topography data (e.g. survey spacing, data density). Also, improved topography is an important contribution, and the impact could be enhanced by providing quantitative information on the improvement or showing comparisons to Bedmap2. There are some statements that seem to conflate active and stable lakes (see comments on lines 37-40), and I believe there could be more discussion on which category the newly discovered lakes fall into. Generally speaking, active lakes identified with satellite observations do not have a clear radar signature, and RES-detected lakes are not observed to have surface elevation changes. The authors hypothesize that these lakes are part of a dynamic drainage system and speculate about cascade-type drainage. It is fine to suggest this, but the fact that many of the lakes in this study are "definite" RES-detected lakes indicates that they could very well fall into the inactive RES lake category. So far, no active drainage has been observed in this region. Previous investigations of SLC and SLE have concluded that these lakes are stable. Perhaps there is a more nuanced stance where RES lakes can be part of a drainage system without the dramatic ice surface drop of active lakes, and the authors do acknowledge that satellite observations of change would be required to confirm drainage. But I think it is important that the authors discuss these contradictory pieces of evidence..*

We appreciate all the elements and resources for improving this manuscript that the referee is offering in this review.

**Specific comments:**

*Line 16: "reported acceleration of ice velocity" Reported an acceleration of ice velocity?*

We will amend the sentence.

*Line 37-40: "These active subglacial lakes have been identified using a range of techniques including satellite measurements of ice surface elevation changes (e.g., Wingham et al., 2006; Smith et al., 2009), characterisation of the subglacial topography from ice surface data (e.g., Bell et al., 2007; Bell, 2008; Jamieson et al., 2016); airborne radio echo sounding (RES) (e.g., Robin et al., 1970; Popov and Masolov, 2003); and/or ground-based RES (e.g., Rivera et al., 2015)." It is unclear what is meant by the identification of lakes through the "characterisation of the subglacial topography from ice surface data." Bell et al. (2007) detected active lakes using satellite data, similarly to Wingham et al. (2006) and Smith et al. (2009). Bell (2008) reviews subglacial lakes gathered from a variety of different sources and surveys, including active lakes detected from satellite data, and non-active lakes detected with radar. The Jamieson et al. (2016) study does not identify lakes. Rather, they hypothesize about potential lake locations by running a water routing model on estimated bed topography. Was this intended to be a statement about active lakes, or subglacial lakes in general? To the best of my knowledge, neither Robin et al. (1970) or Popov and Masolov (2003) have identified active lakes; the lakes they found are considered stable. The Rivera et al. (2015)*

*study also concluded that their lake was stable. The only study that I am aware of that has seen any radiometric evidence for active lakes is Langley et al. (2011): Langley, K., Kohler, J., Matsuoka, K., Sinisalo, A., Scambos, T., Neumann, T., ... & Albert, M. (2011). Recovery Lakes, East Antarctica: Radar assessment of subglacial water extent. Geophysical Research Letters, 38(5).*

We refer to subglacial lakes in general as opposed to just 'active' lakes. We will rewrite this paragraph to clarify how subglacial lakes in general are identified, and also will clarify the methods by which active lakes have been defined. We will check the specific papers as we do this.

*Line 47-49: "Given the fact that this region is located up-ice of the fastest-changing ice streams in the world (e.g., Pine Island Glacier and Thwaites Glacier), and that they are some of the most vulnerable glaciers to ongoing climate change (Martin et al., 2019)..." This statement could be better cited. Some options: Rignot, E., Mouginot, J., Morlighem, M., Seroussi, H., & Scheuchl, B. (2014). Widespread, rapid grounding line retreat of Pine Island, Thwaites, Smith, and Kohler glaciers, West Antarctica, from 1992 to 2011. Geophysical Research Letters, 41(10), 3502-3509. Rignot, E., Mouginot, J., Scheuchl, B., van den Broeke, M., van Wessem, M. J., & Morlighem, M. (2019). Four decades of Antarctic Ice Sheet mass balance from 1979– 2017. Proceedings of the National Academy of Sciences, 116(4), 1095-1103. Joughin, I., Smith, B. E., & Medley, B. (2014). Marine ice sheet collapse potentially under way for the Thwaites Glacier Basin, West Antarctica. Science, 344(6185), 735- 738*

We appreciate the suggested references, and we will include some, or all of them in the text.

*Line 85: The BBAS acronym is not defined Section*

BBAS is the name (not directly an acronym) used to refer to flight lines from the 2004/2005 PASIN survey over PIG (Vaughan et al., 2006). We will be explicit about this in the text.

*2.3: There are two different definitions of specularity. It might be helpful to explain that the method in this manuscript is different from the specularity calculation method in Schroeder et al. (2014) to avoid confusion.*

We will clarify in the text that we use Carter et al. (2007) definition for specularity.

*Line 139: "in a GIS" In a GIS software?*

We will change this in the text complementing GIS with the word software.

*Section 2.5: The methods for surface area and volume calculation are not convincing. The dimensions of two lakes (SLC and SLE) do not provide a statistically robust or representative basis for the dimensions of other lakes. The lake geometry outlined in Figure 3 seems unrealistic, especially if a lake does not happen to fall within a trough or if there is sediment infill. And since the data is not SAR processed, isn't it possible that the hyperbola slopes are different from the actual topography?*

Although, previous studies have made similar assumptions on the shape (i.e., circular shape) in calculating the area for their hypothesised subglacial lakes we do acknowledge the area and volume estimations are subject to very large uncertainties. The other reviewer also made this comment. As a consequence, we will remove this from the text – it will not significantly impact the overall findings of the paper.

*Section 2.6: Given that the water routing model is the primary piece of evidence for the connected drainage hypothesis, I think you can elaborate on the topography. It would be helpful to include information on radar grid spacing, changes from Bedmap2, or percentage of grid cells within 5km of a data point.*

We will make sure we describe the generation for the new DEM fully and that we describe the features within it carefully with an eye on how they end up controlling the drainage and connections between lakes. We will make explicit in the text the references where details on the BBAS survey can be found and will show the unpublished radar survey grid from CECs in a figure. Also, we will show a figure with the differences in the new DEM model (this work) and Bedmap2.

*Lines 255-256: Is the order of magnitude uncertainty in cumulative lake volume really make this a meaningful result, especially given the assumptions in volume calculation?*

We do acknowledge these uncertainties in the different methods applied. Therefore, will remove this estimation from the text.

*Line 302: "the range of length notably smaller" The range of length is notably smaller?*

We will add the word "is".

*Line 310: "poorly constrained" Are poorly constrained?*

We will reword this to point out that Geothermal Heat Flux values are variable depending on the selected technique to model it; and also, that the resolution of the models may not show localized highs in the Heat flux.

*Section 4.2: Is it being hypothesized that these lakes are active lakes? It should be discussed that radar-detected lakes do not have drainage that can be observed from the surface. Or maybe radar-detected lakes are more dynamic than previously expected, but not active enough to be seen from the surface? If you are hypothesizing a more active regime, it might be helpful to cite the MacKie et al. (2020) study which predicts that there are active lakes in the Ellsworth region. MacKie, E. J., Schroeder, D. M., Caers, J., Siegfried, M. R., & Scheidt, C. (2020). Antarctic Topographic Realizations and Geostatistical Modeling Used to Map Subglacial Lakes. Journal of Geophysical Research: Earth Surface.*

Thanks for your suggestion and for the reference. In this article we hypothesize that some of these subglacial lakes may be part of wider active subglacial hydrological drainage system without ice surface changes, provided that hydrological system is in steady state. As long as the rates and locations of flowing water at the base of the ice do not change, it would not affect the surface elevation or they may not be noticed on the surface. We will discuss this more fully in the text and will refer carefully to the MacKie et al (2020) study too.

*Line 335-335: "The identification of any such episodic draining (e.g., Wingham et al., 2006) would require analysis of ice elevation changes to capture water infilling/drainage through time." Following on the comment above, there'll have to be a thoughtful statement about why you think these lakes could be detected by satellite, when so far all other radar-detected lakes (with a possible few exceptions) have not.*

You are correct, very few subglacial lakes detected by radar have also been identified by satellite means. It may be the case that the lake is too small relative to the ice thickness or the recharge period is too long, and the modern satellite have not been able yet to observe one of the drainage events. We remove this statement.

We will change it to downstream.

*423-424: Why is it assumed that the observed lake lengths are the minimum length?*

We made two different assumptions to produce two different ideas of lake size because the radar may pick up at most, the longest dimension of the lake, and at a minimum, it would pick up the shortest dimension of the lake – thus we'd produced two end member estimates (with very large uncertainties). However, we recognise these assumptions have a considerably imprecision (both reviewers commented on this) and therefore we will remove the section of lake dimensions because it does not significantly impact our key findings for the paper.

---

## Referee Report (RR1)

**Review of "Subglacial lakes and hydrology across the Ellsworth Subglacial Highlands, West Antarctica" by Napoleoni et al.**

The revised article addresses the major challenges with the lake volume estimate. The methods section shows improvement in clarity. The revision provides a more nuanced discussion of the potential for dynamic hydrology. Some minor revisions are recommended for further improving clarity and writing quality.

**Specific comments:**

Line 51: "many hypotheses remain untested"

Which hypotheses? A more specific statement would be helpful for improving clarity.

Line 188: "BedMap2" → "Bedmap2"

There are other lines in the text that also need to be corrected to Bedmap2.

Line 198: "since Bedmap2…"

Informal language.

Line 242: "subglacial range" → "subglacial mountain range"?

Line 286: "Figure9c" → "Figure 9c"

Line 304-305: Is this sentence intended to be a single paragraph? Or should it be appended to the paragraph above?

Line 406: "Likely?" or possible

Line 414: "displace … water along this routing" → "displace … water along this flowpath"?

Line 412: "high hydraulic areas" and "low hydraulic areas" → "areas with high hydraulic potential"?

Line 419-420: "... it is possible that under different ice sheet configurations both subglacial lakes were connected hydrologically."

Is it also possible that the lack of connection is from uncertainty in topography or assumptions in the water routing model?

Line 472: "Bedmachine" should be "BedMachine"

Line 475: "Although new and/or more detailed subglacial water or drainage systems could be identified in future RES campaigns, the main drainage pattern would not be substantially different to that which we have identified under the modern ice sheet configuration"

The paragraph after this sentence seems to say that more RES could contribute to significant improvements in hydrological understanding, which seems contradictory to this statement. Line 437 also argues for more surveying.

Figure 7 caption: "ice sheet boundaries" → "catchment boundaries"?

---

## Author Response (AR2)

**Responses to Reviewer 2 for: "Subglacial lakes and hydrology across the Ellsworth Subglacial Highlands, West Antarctica" (MC No: tc-2020-68).**

We appreciate the comments from the referee. We have now made an exhaustive checking on the writing to improve the text and thus providing a further clarity and writing quality. Below we respond (non-highlighted text) to the comments of reviewer 2 (*italic* and highlighted in grey).

**Anonymous referee #2**

*The revised article addresses the major challenges with the lake volume estimate. The methods section shows improvement in clarity. The revision provides a more nuanced discussion of the potential for dynamic hydrology. Some minor revisions are recommended for further improving clarity and writing quality.*

Our responses to the specific comments are detailed below.

**Specific comments:**

*Line 51: "many hypotheses remain untested"*
*Which hypotheses? A more specific statement would be helpful for improving clarity.*

We have provided some examples in the text.

*Line 188: "BedMap2"  "Bedmap2"*
*There are other lines in the text that also need to be corrected to Bedmap2.*

We have now changed the word "BedMap2" for "Bedmap2" throughout the text.

*Line 198: "since Bedmap2. . . "*
*Informal language.*

We have now changed this word.

*Line 242: "subglacial range"  "subglacial mountain range"?*

We have clarified this as suggested.

*Line 286: "Figure9c"  "Figure 9c"*

We have now added the space between Figure and the number.

*Line 304-305: Is this sentence intended to be a single paragraph? Or should it be appended to the paragraph above?*

We have now appended the sentence to the paragraph above.

We have changed "likely" for "possible".

*Line 414: "displace ... water along this routing" "displace ... water along this flowpath"?*

We have now changed "routing" for "flowpath".

*Line 412: "high hydraulic areas" and "low hydraulic areas" "areas with high hydraulic potential"?*

We have now changed this as suggested. Thanks.

*Line 419-420: "... it is possible that under different ice sheet configurations both subglacial lakes were connected hydrologically."*
*Is it also possible that the lack of connection is from uncertainty in topography or assumptions in the water routing model?*

Yes - it could also be related to uncertainty in topography or assumptions in the water routing model. We have now clarified this in the text. Thanks.

We have now changed the word "Bedmachine" for "BedMachine" throughout the text.

We think more RES campaigns may improve our understanding of the subglacial hydrology (e.g. identification of new subglacial lakes) but will not change the main hydrological outlets we have identified in this work. We have now clarified this in the text.

We have now changed for "catchment boundaries".